# SHARP-Distill: A 68× Faster Recommender System with Hypergraph Neural Networks and Language Models

**Saman Forouzandeh** [1]  **Parham Moradi** [1]  **Mahdi Jalili** [1]

## Abstract

This paper proposes SHARP-Distill (**S**peedy **H**ypergraph **A**nd **R**eview-based **P**ersonalised **Distill**ation), a novel knowledge distillation approach based on the teacher-student framework that combines Hypergraph Neural Networks (HGNNs) with language models to enhance recommendation quality while significantly improving inference time. The teacher model leverages HGNNs to generate user and item embeddings from interaction data, capturing high-order and group relationships, and employing a pre-trained language model to extract rich semantic features from textual reviews. We utilise a contrastive learning mechanism to ensure structural consistency between various representations. The student includes a shallow and lightweight GCN called CompactGCN designed to inherit high-order relationships while reducing computational complexity. Extensive experiments on real-world datasets demonstrate that SHARP-Distill achieves up to 68× faster inference time compared to HGNN and 40× faster than LightGCN while maintaining competitive recommendation accuracy.

## 1. Introduction

Recently, GNNs have gained significant success in recommender systems due to their ability to model complex relationships between users and items(Wu et al., 2022). However, despite their advantages, existing GNN-based recommender systems still face several challenges. First, they rely heavily on the user-item interaction matrix, which is often sparse, leading to difficulty in accurately capturing interactions. A possible solution to this issue is utilising multiple sources of information such as social trust(Fan et al., 2019a) or user reviews (Fan et al., 2019b) in the recommendation process.The second challenge is the inference time, as real-world GNN-based methods typically employ deep GNN layers to capture high-order relationships. However, this depth significantly increases computational overhead, leading to longer inference times. While HGNNs (Feng et al., 2019) can capture high-order relationships, they also require a high computational resources during inference time(Yu et al., 2024). Recent approaches like GLNN (Zhang et al., 2021), KRD (Wu et al., 2023), and LightHGNN (Feng et al., 2024) have attempted to address this issue through the teacher-student knowledge distillation approach. Recent approaches, such as GLNN (Zhang et al., 2021), KRD (Wu et al., 2023), and LightHGNN (Feng et al., 2024), have attempted to address this issue using the teacher-student knowledge distillation framework. However, they often struggle to effectively transfer the complex structural information learned by the teacher model. These methods primarily rely on soft labels for knowledge transfer, which fail to capture intricate high-order relationships and structural dependencies. As a result, soft labels alone are insufficient for preserving the rich structural knowledge inherent in hypergraphs. This limitation raises an important question:

**Q1**: How can we design a knowledge transfer methodology that effectively transfers the learned knowledge from a pre-trained HGNN to a lightweight model for recommender systems, while maintaining recommendation accuracy?

**Q2**: How can we efficiently align heterogeneous embeddings derived from different information sources while preserving their unique characteristics?

**Q3:**How can both structural and positional information of complex networks be effectively leveraged in embedding representations?

In this paper, we propose SHARP-Distill, a knowledge distillation framework combining HGNNs and pre-trained language models for high-speed recommender systems. Our method follows a teacher-student strategy, where the teacher model integrates HGNNs to capture high-order user-item interactions and DeBERTa (He et al., 2020a) as a pre-trained

*Equal contribution  [1]School of Engineering, RMIT University, Melbourne, Australia . Correspondence to: Saman Forouzandeh <saman.forouzandeh@rmit.edu.au>.

*Proceedings of the 42nd International Conference on Machine Learning*, Vancouver, Canada. PMLR 267, 2025. Copyright 2025 by the author(s).

language model to extract rich textual representations. We employ contrastive learning (CL) mechanism to effectively align embeddings from multiple sources, such as user-item interactions and user reviews, ensuring a cohesive representation while preserving essential information from each modality. The student model includes a lightweight and single-layer GCN called CompactGCN without non-linear activation, designed for efficiency while retaining high-order dependencies. The student model also employs a contrastive learning (CL) mechanism to align embeddings generated by CompactGCN with those produced by the deep HGNNs in the teacher model. This alignment facilitates the effective transfer of structural and positional knowledge from the teacher to the student, ensuring the preservation of high-order relationships while maintaining computational efficiency (Forouzandeh et al., 2025). This enables Compact-GCN to inherit essential relational knowledge while significantly reducing computational costs. Extensive experiments on real-world datasets show that SHARP-Distill achieves up to **68× faster** inference than HGNN and **40× faster** inference than LightGCN while maintaining or surpassing state-of-the-art recommendation accuracy. Contributions of the Proposed Method are:

1. SHARP-Distill effectively captures both structural patterns from user-item interactions and semantic features from textual reviews.

2. We introduce CompactGCN, a lightweight single-layer GCN without non-linear activations, is specifically designed to inherit high-order relationships from the teacher while significantly reducing computational complexity.

3. Our contrastive learning-based distillation approach integrates structural and positional similarities, ensuring that the student model effectively captures graph topology and node relationships while maintaining computational efficiency.

## 2. Preliminaries

**Graph and Hypergraph:** A graph $G = (V, E)$ consists of a set of nodes $V$ and edges $E$, where each edge $e = (v_i, v_j)$ connects two nodes, $v_i$ and $v_j$. The structure of a graph can be represented by its adjacency matrix $A \in \mathbb{R}^{n \times n}$, where $A_{ij} = 1$ if there is an edge between nodes $v_i$ and $v_j$, and 0 otherwise. This representation is limited to pairwise relationships between nodes. A hypergraph $\mathcal{H} = (\mathcal{V}, \mathcal{E})$, on the other hand, generalises the concept of a graph by allowing hyperedges $e \in \mathcal{E}$ to connect any subset of nodes, enabling the representation of complex, multi-node relationships. A hypergraph is represented by its incidence matrix $\mathcal{H} \in \mathbb{R}^{n \times m}$, where $n$ is the number of nodes, $m$ is the number of hyperedges, and

$\mathcal{H}_{ij} = 1$ if node $v_i$ is connected to hyperedge $]_j$, and 0 otherwise. This matrix effectively captures high-order interactions among nodes, making it suitable for modeling heterogeneous graphs. To convert a hypergraph into a homogeneous graph, we can compute a projected adjacency matrix $A \in \mathbb{R}^{n \times n}$ as $A = \mathcal{H}W\mathcal{H}^\top - D_v$, where $W \in \mathbb{R}^{m \times m}$ is the hyperedge weight matrix, and $D_v$ is the diagonal node degree matrix, with $D_v(i, i) = \sum_{j=1}^{m} \mathcal{H}_{ij}$. This transformation reduces the hypergraph to a traditional graph where two nodes are connected if they share common hyperedges.

**Hypergraph Neural Network (HGNN)** is a powerful extension of the GCN designed to capture high-order relationships among nodes by leveraging the structure of hypergraphs. HGNNs use the hypergraph Laplacian, derived from the node degree matrix $D_v$ and the hyperedge degree matrix $D_e$, to propagate information across the hypergraph. This is achieved through a message-passing mechanism that updates node features using the formula:

$$H^{(l+1)} = \sigma \left( D_v^{-1/2} \mathcal{H} W D_e^{-1} \mathcal{H}^T D_v^{-1/2} H^{(l)} \Theta^{(l)} \right) \quad (1)$$

where $\Theta^{(l)}$ represents the learnable weights and $\sigma(\cdot)$ is a non-linear activation function. This formulation allows HGNNs to effectively aggregate and propagate features, accounting for the structure of the hypergraph, thus enabling the modeling of complex dependencies between nodes. HGNNs have demonstrated their versatility and effectiveness in various applications, including recommendation systems, social networks, and biological data analysis (Feng et al., 2019).

**DeBERTa** (Decoding-enhanced BERT with Disentangled Attention) (He et al., 2020a) is a pre-trained language model that extracts contextual embeddings from text through disentangled attention mechanisms. For each user-item pair $(u_i, v_j)$ with an associated review $r_{ij}$, we process the text through DeBERTa to obtain contextual representations, which we denote as $X' = \text{DeBERTa}(X)$ where $X'$ represents the output embedding after passing the input text $X$ through the model.

## 3. Methodology

We propose a novel teacher-student knowledge distillation framework for a high-speed recommender system. As illustrated in Figure 1, our framework integrates HGNNs with a pre-trained language model (PLM) to effectively capture high-order relations and textual semantics. By integrating multiple information sources, such as user-item interactions and user reviews, our framework aims to address challenges including data sparsity, cold-start problems, and noisy interactions, thereby ensuring more accurate and efficient recommendations. Additionally, our framework transfers knowledge to a lightweight student model, significantly en-

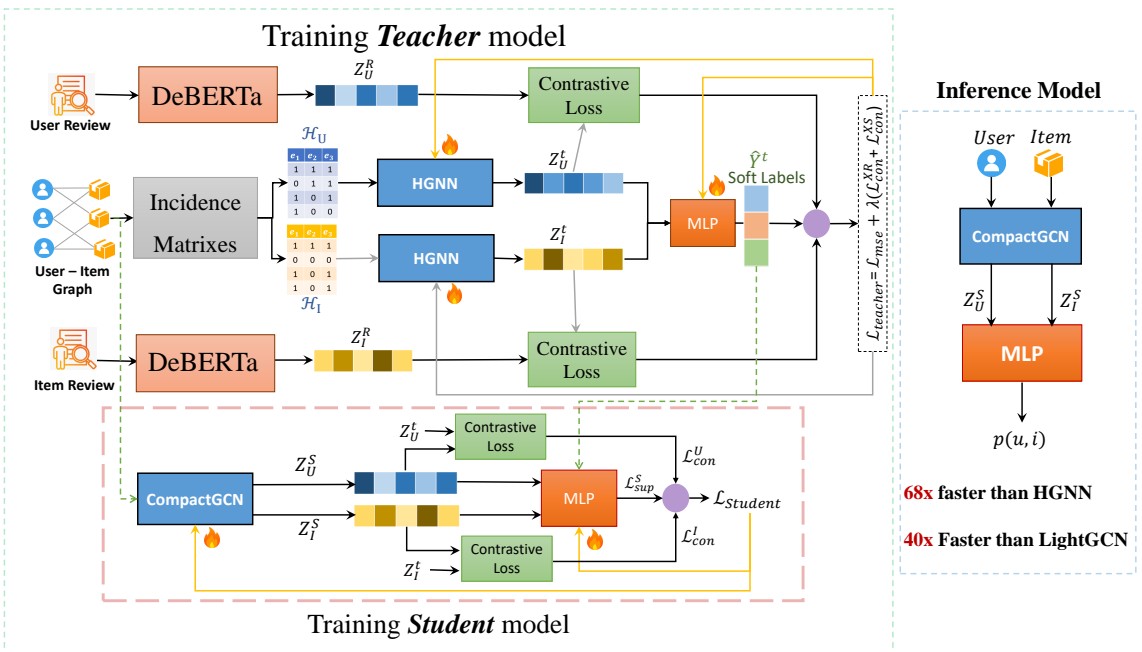

*Figure 1.* SHARP-Distill framework architecture. The Teacher model integrates dual Hypergraph Neural Networks (HGNNs) with DeBERTa to capture high-order relations in user-item interactions and semantic features from textual reviews through contrastive learning. The Student model employs CompactGCN enhanced with contrastive learning to align its embeddings with teacher representations. Knowledge distillation combines structural and positional knowledge transfer, achieving 68× and 40× faster inference times compared to HGNN and LightGCN, respectively.

hancing inference speed while maintaining recommendation accuracy. The details of each component are described in their respective subsections.

**Teacher Model:** The teacher model integrates three key components: two HGNNs for modeling high-order user-item interactions, DeBERTa as a pre-trained language model for extracting semantic features from textual reviews, and a Multi-Layer Perceptron (MLP) for predicting ratings. The HGNNs capture high-order relations in the user-item interaction graph using the hypergraph Laplacian. Let us define the user incidence matrix as $\mathcal{H}_U \in \mathbb{R}^{n \times m}$, where $n$ denotes the number of users and $m$ represents hyperedges (items). The normalised hypergraph Laplacian is defined as:

$$L = D_v^{-1/2} \mathcal{H}_U W D_e^{-1} \mathcal{H}_U^T D_v^{-1/2}, \quad (2)$$

where $D_v \in \mathbb{R}^{n \times n}$ and $D_e \in \mathbb{R}^{m \times m}$ are diagonal degree matrices for vertices and hyperedges, and $W \in \mathbb{R}^{m \times m}$ is the hyperedge weight matrix. The message propagation at each layer $l$ follows:

$$H^{(l+1)} = \sigma\left(D_v^{-1/2} \mathcal{H}_U W D_e^{-1} \mathcal{H}_U^T D_v^{-1/2} H^{(l)} \Theta^{(l)}\right), \quad (3)$$

where $\sigma(\cdot)$ is the ReLU activation function, and $\Theta^{(l)} \in \mathbb{R}^{d \times d}$ is the trainable weight matrix at layer $l$. The final user and item embeddings after $L$ layers are $Z_U^t = H^{(L)} \in \mathbb{R}^{n \times d}$ and $Z_I^t = H^{(L)} \in \mathbb{R}^{m \times d}$ respectively.

To enhance alignment between textual embeddings from De-BERTa ($Z_U^R$, $Z_I^R$) and structural embeddings from HGNNs ($Z_U^t$, $Z_I^t$), we employ cross-modal and intra-domain contrastive learning. The cross-modal contrastive loss aligns HGNN and DeBERTa embeddings:

$$L_{\text{con}}^{XR} = -\frac{1}{|S_X|} \sum_{i=1}^{|S_X|} \log \frac{\exp(\text{sim}(Z_X^t[i], Z_X^R[i])/\tau)}{\sum_{j=1}^{|S_X|} \exp(\text{sim}(Z_X^t[i], Z_X^R[j])/\tau)}, \quad (4)$$

where $X \in \{U, I\}$, $\text{sim}(\cdot, \cdot)$ is cosine similarity, and $\tau$ is a temperature parameter. The intra-domain contrastive loss is:

$$L_{\text{con}}^{XS} = -\frac{1}{|S_X|} \sum_{i=1}^{|S_X|} \log \frac{\exp(\text{sim}(Z_X^t[i], Z_X^t[j])/\tau)}{\sum_{k=1}^{|S_X|} \exp(\text{sim}(Z_X^t[i], Z_X^t[k])/\tau)}, \quad (5)$$

where $S_U$ and $S_I$ denote user and item sets, respectively. After contrastive alignment, the HGNN embeddings and textual embeddings from DeBERTa are fused via a $k$-layer MLP:

$$h_1 = \sigma([Z_U^t[i] \| Z_I^t[j] \| Z_U^R[i] \| Z_I^R[j]] W_1 + b_1), \quad (6)$$

$$h_l = \sigma(h_{l-1} W_l + b_l), \quad l \in \{2, \ldots, k-1\}, \quad (7)$$

where $\|$ denotes concatenation, $W_l$ and $b_l$ are trainable parameters, and $\sigma(\cdot)$ is ReLU activation. The output layer

of the MLP generates rating predictions:

$$\hat{Y}_{ij}^t = \sigma_{\text{out}}(h_k W_k + b_k), \qquad (8)$$

where $\sigma_{\text{out}}(\cdot)$ is the sigmoid activation for rating prediction. The supervised loss component is the Mean Squared Error (MSE) loss:

$$L_{\text{mse}} = \frac{1}{|Y|} \sum_{(i,j) \in Y} (Y_{ij} - \hat{Y}_{ij}^t)^2, \qquad (9)$$

where $Y_{ij}$ is the ground truth rating. The final teacher objective combines supervised and contrastive components:

$$L_{\text{teacher}} = L_{\text{mse}} + \lambda \sum_{X \in \{U,I\}} (L_{\text{con}}^{XR} + L_{\text{con}}^{XS}), \qquad (10)$$

where $\lambda$ is a trade-off parameter balancing the objectives. The teacher's predictions are then converted to soft labels using temperature scaling for knowledge distillation:

$$Y^t = \text{softmax}(\hat{Y}^t/T) = \frac{\exp(\hat{Y}^t/T)}{\sum_j \exp(\hat{Y}_j^t/T)}, \qquad (11)$$

where $T > 1$ is the temperature parameter following an annealing schedule:

$$T(e) = T_{\text{max}} - (T_{\text{max}} - T_{\text{min}}) \cdot \min\left(1, \frac{e}{E}\right). \qquad (12)$$

For the student model, we employ a crucial embedding interpolation mechanism that facilitates effective knowledge transfer. This interpolation combines the independently learned student embeddings with the well-trained teacher embeddings:

$$\begin{aligned} Z_U^s &= \gamma Z^s[:\mathcal{N}_u, :] + (1 - \gamma)Z_U^t, \\ Z_I^s &= \gamma Z^s[\mathcal{N}_u :, :] + (1 - \gamma)Z_I^t, \end{aligned} \qquad (13)$$

The interpolation coefficient $\gamma \in [0, 1]$ controls the balance between student and teacher embeddings. $Z^s[:\mathcal{N}_u, :]$ and $Z^s[\mathcal{N}_u :, :]$ represent the student's user and item embeddings, while $Z_U^t$ and $Z_I^t$ are the teacher's embeddings. When $\gamma$ is near 0, the student relies on the teacher's knowledge, which is useful early in training. As $\gamma$ increases, the student transitions to its own learned representations.

**Student Model:** The student model consists of a lightweight GCN called CompactGCN along with an MLP. CompactGCN is specifically designed to efficiently inherit essential structural and topological information from the teacher model, ensuring a balance between computational efficiency and recommendation accuracy. Let $X \in \mathbb{R}^{n \times d}$ be the feature matrix of users/items, and $A^s \in \mathbb{R}^{n \times n}$ represent direct user-item interactions. The normalised Laplacian matrix is defined as $\hat{A}^s = D^{-1/2}(A^s + I)D^{-1/2}$, where

$D \in \mathbb{R}^{n \times n}$ is the degree matrix and $I$ is the identity matrix. The initial node embeddings are computed as:

$$Z^s = \hat{A}^s X W^s, \qquad (14)$$

where $W^s \in \mathbb{R}^{d \times h}$ is the transformation matrix for structural features, $h$ denotes the hidden dimension of the output embeddings, and $Z^s \in \mathbb{R}^{n \times h}$ represents the learned node embeddings. Following Hinton et al. (Hinton, 2015), the supervised loss combines hard and soft targets:

$$L_{\text{sup}}^s = (1 - \beta)L_{\text{hard}} + \beta L_{\text{soft}}, \qquad (15)$$

where $\beta \in [0, 1]$ balances the contributions of hard and soft targets. The two loss components are:

$$\begin{aligned} L_{\text{hard}} &= \frac{1}{|Y|} \sum_{(i,j) \in Y} (Y_{ij} - \hat{Y}_{ij}^s)^2, \\ L_{\text{soft}} &= T^2 \cdot \text{KL}(Y^t/T \parallel \hat{Y}^s/T), \end{aligned} \qquad (16)$$

where the $T^2$ factor ensures proper gradient scaling (Phuong & Lampert, 2019).

**Contrastive Learning for Knowledge Transfer:** Traditional knowledge distillation methods primarily rely on soft labels, which are insufficient for capturing the rich structural information inherent in hypergraphs and the complex multi-modal relationships between user-item interactions and textual reviews. To address this limitation, our approach employs contrastive learning to align representations between teacher and student models, offering several key advantages: (1) *Structural preservation*: Unlike soft labels that only provide prediction-level guidance, contrastive learning explicitly preserves the relational structure and topological properties learned by the teacher's HGNNs; (2) *Multi-modal alignment*: It effectively bridges the semantic gap between different representation spaces (hypergraph embeddings vs. lightweight GCN embeddings); (3) *Robustness to noise*: Contrastive learning is more robust to noisy interactions compared to direct embedding matching, as it focuses on relative similarities rather than absolute values.

To effectively transfer knowledge, the student model leverages contrastive learning incorporating both embedding-based similarity and hypergraph-informed topological similarity. The hypergraph positional encoding captures the teacher's topological knowledge:

$$P_u = \sum_{k=1}^{K} \beta^k \left( D_v^{-1/2} \mathcal{H}_U W D_e^{-1} \mathcal{H}_U^T D_v^{-1/2} \right)^k e_u, \qquad (17)$$

where $e_u$ is the one-hot vector for node $u$, $K$ denotes the maximum propagation steps, and $\beta \in (0, 1)$ is a decay factor. This formulation captures both local and higher-order dependencies in the hypergraph topology. We define

a comprehensive similarity function that integrates both embedding-based and positional similarities:

$$\mathcal{S}(u,v) = \alpha \left( \frac{Z_u^t \cdot Z_v^s}{\|Z_u^t\|\|Z_v^s\|} \right) + (1-\alpha) \left( \frac{P_u \cdot P_v}{\|P_u\|\|P_v\|} \right), \quad (18)$$

where $Z_u^t$ and $Z_v^s$ are the embeddings from the teacher and student models, respectively. $P_u$ and $P_v$ represent the hypergraph positional encodings, and $\alpha \in [0,1]$ controls the trade-off between embedding similarity and positional similarity. The contrastive learning objective aligns user and item representations between the teacher and student models. For user alignment:

$$L_{\text{con}}^U = -\frac{1}{|U|} \sum_{u \in U} \log \frac{\exp(\mathcal{S}(u,u)/\tau)}{\sum_{v \in U} \exp(\mathcal{S}(u,v)/\tau)}, \quad (19)$$

and for item alignment:

$$L_{\text{con}}^I = -\frac{1}{|I|} \sum_{i \in I} \log \frac{\exp(\mathcal{S}(i,i)/\tau)}{\sum_{j \in I} \exp(\mathcal{S}(i,j)/\tau)}, \quad (20)$$

where $U$ and $I$ represent the sets of all users and items, respectively. The temperature parameter $\tau > 0$ controls the concentration of the similarity distribution. Here, $\mathcal{S}(u,u)$ denotes the similarity between a user's representations in the teacher and student models, while $\mathcal{S}(u,v)$ measures the similarity between different users. The final student loss integrates supervised learning, knowledge distillation, and contrastive components:

$$L_{\text{student}} = L_{\text{sup}}^s + \lambda_1 L_{\text{con}}^U + \lambda_2 L_{\text{con}}^I, \quad (21)$$

where $L_{\text{sup}}^s$ combines hard and soft targets, and hyperparameters $\lambda_1, \lambda_2 > 0$ control the contributions of user and item contrastive learning objectives. This formulation facilitates effective knowledge transfer by explicitly aligning corresponding user and item representations between teacher and student models while preserving their relative relationships in the embedding space. The framework enables the student model to inherit both embedding-based and hypergraph-informed topological knowledge, ensuring efficient representation learning through CompactGCN while maintaining the structural richness of the teacher model. Theoretical proofs and supporting lemmas for the model are provided in Appendix A. The complete training procedure is detailed in Algorithm 1 in Appendix B.

## 4. Experiments

In our experiments, we utilise five well-known datasets. First, we use the Yelp Tips dataset[1]. Additionally, we leverage the Amazon datasets, including four more datasets, covering different domains within the Amazon collection—

CDs, Cellphones, Beauty, and Sports [2]—are used as well, as referenced in (Zhang et al., 2024). All of these datasets contain user interactions, items, and user reviews of the items. In this research, we construct two hypergraphs: one for users and one for items. Based on these, we generate two sets of review-based embeddings—one for users and one for items—by processing the reviews submitted by users and the reviews received by items from various users. A summary of the dataset statistics is provided in Table 6 in the Appendix C. We evaluate the proposed method based on the generated recommendations in two key aspects: accuracy and inference time. For accuracy, we use the metrics Precision@k (P@k), Recall@k (R@k), F1-Score (F@k), and NDCG@k (N@k), where $k = 10$ is used during the testing phase. Table 1 presents the accuracy results across the five datasets. Each experiment was conducted 10 times using different random seeds, and we report both the average performance and standard deviation. For each dataset, 20% of the data was used for validation and 10% for testing. We evaluate SHARP-Distill using four distinct methodological approaches to provide comprehensive performance analysis.

**Hypergraph-based Methods:** We benchmark SHARP-Distill against state-of-the-art hypergraph neural networks: LightGCN (He et al., 2020b); Hypergraph Contrastive Collaborative Filtering (HCCF) (Xia et al., 2022); and Hypergraph Attention (HGAtt) (Bai et al., 2021). We also include the Graph-centric Contrastive Framework for Graph Matching (GCGM) (Bo & Fang, 2024), which exemplifies recent advances in contrastive learning on graph structures. **Large Language Model-based Methods:** We evaluate two LLM-enhanced recommendation approaches: SAID (Hu et al., 2024), which integrates semantic knowledge from large language models, and Prompt Distillation (POD) (Li et al., 2023), which transfers knowledge via prompt-based learning. **Knowledge Distillation Methods:** We compare against teacher-to-student distillation frameworks that leverage hypergraph structures: KRD (Wu et al., 2023) and LightHGNN (HGNN-to-MLP) (Feng et al., 2024). Table 1 presents a comprehensive evaluation across five real-world datasets, demonstrating SHARP-Distill's superior performance on multiple metrics.

SHARP-Distill demonstrates superior performance across most metrics and datasets, achieving the best results in 11 out of 15 metric-dataset combinations. Particularly notable improvements are observed in the Sports dataset, where SHARP-Distill significantly outperforms all baselines across all metrics. The method shows strong competitive performance against LLM-based approaches, with SAID achieving best performance in Cellphones P@10 (7.83) and sharing best performance with HGAtt in CDs N@10 (12.24), demonstrating the effectiveness of semantic knowledge in-

---

[1] https://www.yelp.com/dataset

[2] http://jmcauley.ucsd.edu/data/amazon/

*Table 1.* Comprehensive experimental results on five real-world datasets. All metrics are presented as percentages (%) with standard deviations. Bold indicates best performance in each category.

| Dataset | Metric | Hypergraph Methods | | | | LLM Methods | | Distillation Methods | | SHARP-Distill |
|---------|--------|----------|------|-------|------|------|-----|-----|----------|---------------|
| | | LightGCN | HCCF | HGAtt | GCGM | SAID | POD | KRD | LightHGNN | |
| **Yelp** | P@10 | $2.27_{\pm 0.032}$ | $3.52_{\pm 0.075}$ | $3.32_{\pm 0.038}$ | $3.45_{\pm 0.042}$ | $3.15_{\pm 0.062}$ | $3.28_{\pm 0.055}$ | $3.11_{\pm 0.059}$ | $3.27_{\pm 0.084}$ | $3.88_{\pm 0.047}$ |
| | R@10 | $1.87_{\pm 0.069}$ | $2.96_{\pm 0.064}$ | $2.53_{\pm 0.064}$ | $2.84_{\pm 0.058}$ | $2.42_{\pm 0.078}$ | $2.65_{\pm 0.071}$ | $2.41_{\pm 0.072}$ | $2.17_{\pm 0.075}$ | $2.75_{\pm 0.065}$ |
| | N@10 | $1.26_{\pm 0.051}$ | $2.25_{\pm 0.069}$ | $2.07_{\pm 0.051}$ | $2.15_{\pm 0.047}$ | $1.95_{\pm 0.067}$ | $2.18_{\pm 0.059}$ | $1.73_{\pm 0.080}$ | $1.74_{\pm 0.095}$ | $2.37_{\pm 0.049}$ |
| **CDs** | P@10 | $11.67_{\pm 0.086}$ | $13.96_{\pm 0.079}$ | $13.21_{\pm 0.125}$ | $13.54_{\pm 0.095}$ | $13.40_{\pm 0.082}$ | $12.88_{\pm 0.091}$ | $12.17_{\pm 0.132}$ | $12.40_{\pm 0.115}$ | $13.75_{\pm 0.076}$ |
| | R@10 | $10.14_{\pm 0.115}$ | $12.05_{\pm 0.093}$ | $12.47_{\pm 0.091}$ | $12.65_{\pm 0.088}$ | $12.65_{\pm 0.085}$ | $12.20_{\pm 0.094}$ | $11.56_{\pm 0.105}$ | $12.54_{\pm 0.087}$ | $13.06_{\pm 0.062}$ |
| | N@10 | $9.75_{\pm 0.079}$ | $11.70_{\pm 0.088}$ | $12.24_{\pm 0.101}$ | $12.08_{\pm 0.094}$ | $12.24_{\pm 0.089}$ | $11.88_{\pm 0.096}$ | $10.77_{\pm 0.146}$ | $11.70_{\pm 0.112}$ | $12.17_{\pm 0.098}$ |
| **Cellphones** | P@10 | $6.15_{\pm 0.034}$ | $7.68_{\pm 0.021}$ | $7.79_{\pm 0.037}$ | $7.82_{\pm 0.032}$ | $7.83_{\pm 0.045}$ | $7.62_{\pm 0.041}$ | $6.68_{\pm 0.058}$ | $6.77_{\pm 0.070}$ | $7.54_{\pm 0.053}$ |
| | R@10 | $4.27_{\pm 0.055}$ | $5.58_{\pm 0.028}$ | $5.22_{\pm 0.036}$ | $5.35_{\pm 0.041}$ | $5.72_{\pm 0.039}$ | $5.60_{\pm 0.044}$ | $4.62_{\pm 0.079}$ | $4.88_{\pm 0.066}$ | $5.77_{\pm 0.044}$ |
| | N@10 | $3.66_{\pm 0.048}$ | $4.30_{\pm 0.037}$ | $4.67_{\pm 0.061}$ | $4.58_{\pm 0.057}$ | $4.69_{\pm 0.052}$ | $4.62_{\pm 0.055}$ | $3.89_{\pm 0.086}$ | $3.95_{\pm 0.072}$ | $4.77_{\pm 0.065}$ |
| **Beauty** | P@10 | $4.27_{\pm 0.067}$ | $5.63_{\pm 0.018}$ | $6.05_{\pm 0.025}$ | $5.95_{\pm 0.028}$ | $6.58_{\pm 0.035}$ | $6.60_{\pm 0.031}$ | $5.53_{\pm 0.057}$ | $6.01_{\pm 0.048}$ | $6.97_{\pm 0.032}$ |
| | R@10 | $3.35_{\pm 0.082}$ | $3.85_{\pm 0.075}$ | $4.62_{\pm 0.069}$ | $4.45_{\pm 0.065}$ | $4.38_{\pm 0.071}$ | $4.29_{\pm 0.076}$ | $4.01_{\pm 0.115}$ | $4.08_{\pm 0.104}$ | $4.52_{\pm 0.089}$ |
| | N@10 | $2.93_{\pm 0.044}$ | $3.38_{\pm 0.048}$ | $3.76_{\pm 0.036}$ | $3.65_{\pm 0.042}$ | $3.97_{\pm 0.058}$ | $4.01_{\pm 0.051}$ | $3.41_{\pm 0.072}$ | $3.55_{\pm 0.083}$ | $4.15_{\pm 0.074}$ |
| **Sports** | P@10 | $2.57_{\pm 0.031}$ | $3.15_{\pm 0.048}$ | $3.56_{\pm 0.042}$ | $3.48_{\pm 0.038}$ | $3.22_{\pm 0.055}$ | $3.41_{\pm 0.049}$ | $3.36_{\pm 0.045}$ | $3.42_{\pm 0.052}$ | $4.27_{\pm 0.024}$ |
| | R@10 | $1.83_{\pm 0.052}$ | $2.42_{\pm 0.031}$ | $2.71_{\pm 0.023}$ | $2.65_{\pm 0.027}$ | $2.35_{\pm 0.048}$ | $2.58_{\pm 0.037}$ | $2.02_{\pm 0.040}$ | $2.23_{\pm 0.038}$ | $3.63_{\pm 0.019}$ |
| | N@10 | $1.10_{\pm 0.047}$ | $1.55_{\pm 0.018}$ | $1.74_{\pm 0.028}$ | $1.68_{\pm 0.032}$ | $1.48_{\pm 0.042}$ | $1.67_{\pm 0.035}$ | $1.48_{\pm 0.036}$ | $1.64_{\pm 0.027}$ | $3.24_{\pm 0.017}$ |

tegration. However, SHARP-Distill maintains dominant performance across recall and NDCG metrics, excelling in 7 out of 10 recall/NDCG combinations. The results validate the effectiveness of our multi-modal knowledge distillation approach, which successfully combines hypergraph neural networks with pre-trained language models while maintaining computational efficiency through the lightweight CompactGCN architecture. Also, a comprehensive comparison of SHARP-Distill with distillation-based recommendation methods is presented in Appendix D.

## 4.1. Hit Ratio

Hit Ratio (HR) is a metric used to evaluate the effectiveness of a recommendation system (Steck, 2011). It measures how often the true, relevant item is included in the Top-K recommended items for each user. For a given dataset, HR@K indicates the proportion of users for whom the correct item appears within the top K recommendations. A higher HR value implies that the model is more successful at making relevant recommendations within the top-K list. In this section, we present the experimental results of our proposed method, **SHARP-Distill**, compared to baseline methods on the two datasets: Yelp and CDs. The Hit Ratio results for various K values (HR@10, HR@20, and HR@50) are shown in Table 2 as percentages (%).

The table shows Hit Ratio (HR) results for the Yelp and CDs datasets at various K values (HR@10, HR@20, HR@50). **SHARP-Distill** demonstrates superior performance across most metrics, achieving the highest HR in 5 out of 6 metric-dataset combinations. For Yelp, **SHARP-Distill** achieves HR@10 of 44.67%, HR@20 of 61.31%, and HR@50 of

84.60%, consistently outperforming all baseline methods. For the CDs dataset, it reaches HR@10 of 54.42% (second to SAID's 51.85%), HR@20 of 72.17%, and HR@50 of 89.48%. The LLM-based methods (SAID and POD) show competitive performance, with SAID achieving the best HR@10 performance on the CDs dataset, demonstrating the effectiveness of semantic knowledge integration. These results highlight **SHARP-Distill**'s superior performance in recommending relevant items, particularly excelling at higher K values where it consistently achieves the best results across both datasets.

## 4.2. Balancing Precision and Inference Time

In recommender systems, balancing precision and inference time is crucial, especially for real-world applications where recommendations must be both accurate and efficient. We evaluate models, including **SHARP-Distill**, on two datasets—Amazon Cellphones and CDs—using precision at top-10 recommendations (P@10) and inference time. While high precision is essential, models like GNNs and HGNNs tend to have longer inference times, posing challenges in time-sensitive applications. Faster models, like MLPs, often sacrifice precision. **SHARP-Distill** leverages knowledge distillation to balance high precision with efficient inference, making it ideal for practical recommender systems. Results in Figure 4 show how each model compares in precision and inference time, highlighting their suitability for different use cases.

**SHARP-Distill** balances precision and inference time effectively across both the Amazon Beauty and Amazon Cellphones datasets, outperforming other models in effi-

*Table 2.* Experimental results on two datasets based on Hit Ratio (HR), presented as percentages (%). Bold indicates best performance.

| Dataset | Metric | Hypergraph Methods | | | | LLM Methods | | Distillation Methods | | SHARP-Distill |
|---|---|---|---|---|---|---|---|---|---|---|
| | | LightGCN | HCCF | HGAtt | GCGM | SAID | POD | KRD | LightHGNN | |
| **Yelp** | HR@10 | 33.65 | 37.47 | 39.15 | 38.45 | 36.85 | 37.92 | 34.58 | 41.34 | 44.67 |
| | HR@20 | 47.71 | 49.35 | 48.19 | 48.85 | 48.15 | 49.28 | 47.34 | 49.64 | 61.31 |
| | HR@50 | 65.18 | 70.22 | 73.43 | 72.18 | 71.45 | 72.85 | 72.05 | 71.39 | 84.60 |
| **CDs** | HR@10 | 42.59 | 49.72 | 48.55 | 49.12 | 51.85 | 50.42 | 43.12 | 46.34 | 54.42 |
| | HR@20 | 56.35 | 62.05 | 64.20 | 63.45 | 65.72 | 64.88 | 60.96 | 63.44 | 72.17 |
| | HR@50 | 70.25 | 79.68 | 83.35 | 81.92 | 85.15 | 84.25 | 73.27 | 75.32 | 89.48 |

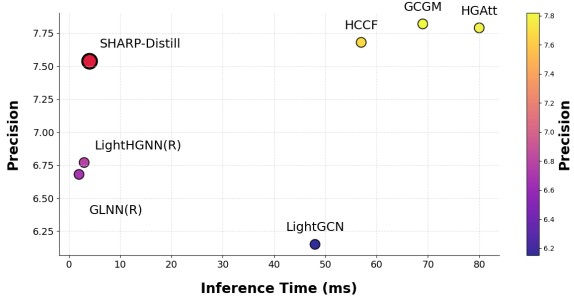

*Figure 2.* Amazon Cellphones (Precision vs. Inference Time)

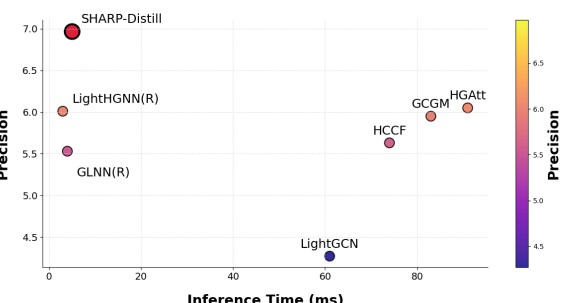

*Figure 3.* Amazon Beauty (Precision vs. Inference Time)

*Figure 4.* Comparison of Models based on Accuracy and Inference Time

ciency without sacrificing accuracy. On Amazon Beauty, it achieves high precision (6.97) with a competitive inference time of 5.0 ms, slightly higher than the fastest model, LightHGNN(R) at 3.0 ms, but with significantly better precision (6.97 vs 6.01). On Amazon Cellphones, **SHARP-Distill** achieves strong precision (7.54), close to HGAtt (7.79), but with much faster inference times (4.0 ms vs 80.0 ms). These results demonstrate **SHARP-Distill**'s ability to deliver accurate recommendations quickly, making it ideal for real-world recommender systems.

In the next step, we evaluate the practical applicability of recommendation systems by analysing computational efficiency during inference, particularly for large-scale applications. This section compares the efficiency of SHARP-Distill with state-of-the-art models (LightGCN and HGNN) across different data scales. Experiments are conducted on two benchmark datasets, Amazon CDs and Yelp, with varying node counts (20K, 40K, 80K, and full dataset size) to assess scalability. We measured the average inference time over 100 runs, excluding data loading time. Results and speed improvements are shown in Table 3.

The experimental results show that SHARP-Distill significantly outperforms baseline models in inference efficiency. On the Amazon CDs dataset, SHARP-Distill achieves up to 40× faster inference than LightGCN and 68× faster than

HGNN at the full dataset scale (136,701 nodes), with increasing speed advantages as network size grows. A similar trend is observed in the Yelp dataset, where SHARP-Distill is 39× faster than LightGCN and 63× faster than HGNN at the full scale (117,302 nodes). These improvements indicate robust, generalisable efficiency across different recommendation scenarios. Unlike the baseline models, which show linear or super-linear growth in inference time, SHARP-Distill exhibits modest increases, thanks to its lightweight architecture, efficient neighbour representation, and optimized model structure.

### 4.3. Ablation Study

In this section, we present findings that assess the effectiveness of soft labels and knowledge transfer methods, with **SHARP-Distill** incorporating both techniques. In contrast, previous research has primarily focused on using soft labels alone for knowledge transfer to the student model (Zhang et al., 2021; Wu et al., 2023; Feng et al., 2024). We evaluate the model based on four configurations: (a) soft labels only, (b) structural knowledge of CL, (c) positional knowledge of CL, and (d) SHARP-Distill. The results, shown in Figure 7, include precision and inference time metrics for each method.

The analysis demonstrates that **SHARP-Distill**, which com-

*Table 3.* Comparison of Inference Time (ms) and Speed Factor Analysis for Amazon CDs and Yelp Datasets

| Dataset | Node Count | LightGCN | HGNN | SHARP-Distill | Times Faster | |
|---|---|---|---|---|---|---|
| | | | | | vs LightGCN | vs HGNN |
| **Amazon CDs** | 20,000 | 45.23 | 68.45 | 2.15 | 21× | 32× |
| | 40,000 | 98.45 | 155.67 | 4.18 | 24× | 37× |
| | 80,000 | 196.78 | 362.34 | 7.62 | 26× | 48× |
| | 136,701 | 395.45 | 668.23 | 9.77 | 40× | 68× |
| **Yelp** | 20,000 | 42.67 | 64.23 | 2.98 | 14× | 22× |
| | 40,000 | 94.23 | 148.67 | 4.74 | 20× | 31× |
| | 80,000 | 188.45 | 277.45 | 7.37 | 26× | 38× |
| | 117,302 | 342.67 | 552.34 | 8.79 | 39× | 63× |

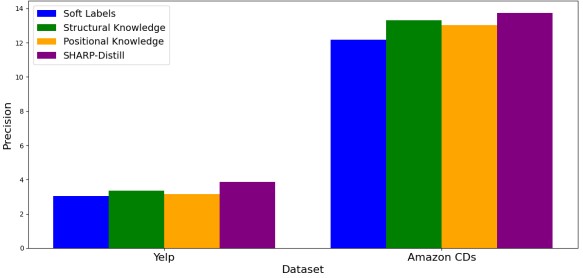

*Figure 5.* Precision-Based Comparison

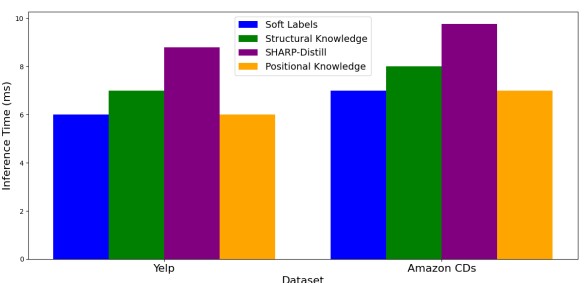

*Figure 6.* Inference Time-Based Comparison

*Figure 7.* The model was evaluated using three types of knowledge transfer methods

bines soft labels with structural knowledge, achieves the highest precision across both datasets. While soft labels alone enable faster inference, **SHARP-Distill** delivers significantly better precision with only a marginal increase in computation time, making the trade-off highly favourable. The model effectively balances accuracy and inference efficiency. To evaluate the contributions of different components in the **SHARP-Distill** model, we conduct an ablation study by selectively removing DeBERTa embeddings ($DeB$) and contrastive learning ($CL$) from the full model. Table 4 summaries the results, showcasing the impact of these components on Precision at 10 (P@10), Recall at 10 (R@10), and Normalised Discounted Cumulative Gain at 10 (N@10), along with the percentage performance drop compared to the full model.

The results clearly highlight the pivotal role of DeBERTa embeddings and contrastive learning in the **SHARP-Distill** model.

**Representation Alignment Evaluation**

We evaluate how effectively the student model captures both semantic and structural knowledge by computing **Centered Kernel Alignment (CKA)** scores between teacher and student embeddings across five datasets. CKA measures representational similarity while being invariant to orthogonal transformations, making it ideal for comparing learned

feature spaces. We analyze three embedding types: HGNN-based structural embeddings (structure only), DeBERTa-based semantic embeddings (text only), and SHARP-Distill embeddings (after contrastive alignment).

As shown in Figure 8, SHARP-Distill consistently achieves the highest CKA alignment with the full teacher model across all datasets, demonstrating superior fusion of semantic and structural modalities compared to individual components. This validates our dual contrastive learning design and confirms effective knowledge transfer from the complex teacher to the lightweight student model.

Table 5 presents comprehensive training and inference analysis, demonstrating SHARP-Distill's efficiency advantages. While requiring slightly more training time than Light-GCN due to the teacher-student framework, SHARP-Distill achieves dramatic inference acceleration (up to 68× faster than the teacher HGNN) with significantly fewer parameters (0.5M vs. 145M). Crucially, only the compact student model is deployed for inference, achieving $\mathcal{O}(|E|d)$ complexity and enabling real-time recommendations.

To gain deeper insights into its components, we conduct several evaluations: DeBERTa's performance is analysed in Appendix E, hyperparameter sensitivity is assessed in Appendix F, layer configurations are examined in Appendix G, training size settings are evaluated in Appendix H, train-

*Table 4.* Impact of Removing Components from the Model

| Model Variation | P@10 | R@10 | N@10 | Drop (%) |
|---|---|---|---|---|
| **Yelp Dataset** | | | | |
| **SHARP-Distill** | 3.88 | 2.75 | 2.37 | - |
| SHARP$-(DeB)$ | 3.15 | 2.21 | 1.85 | -20.13 |
| SHARP$-(CL)$ | 2.74 | 1.93 | 1.46 | -32.35 |
| **Amazon CDs Dataset** | | | | |
| **SHARP-Distill** | 13.75 | 13.06 | 12.17 | - |
| SHARP$-(DeB)$ | 12.84 | 12.51 | 11.43 | -5.62 |
| SHARP$-(CL)$ | 12.35 | 12.18 | 11.13 | -8.49 |

*Table 5.* Training and Inference Analysis for SHARP-Distill and Baselines

| Dataset | Model | Train Time (hrs) | Infer Time (ms) | Params | Comp. (Inference) | Deployed? |
|---|---|---|---|---|---|---|
| Amazon–CDs | HGNN + DeBERTa (Teacher) | 4.2 | 668.23 | 145M | $\mathcal{O}(n^2 d + md)$ | ✗ |
| | LightGCN | 2.0 | 395.45 | 1.5M | $\mathcal{O}(\|E\|d)$ | ✓ |
| | SHARP-Distill (Student) | 0.3 | **9.77** | **0.5M** | $\mathcal{O}(\|E\|d)$ | ✓ |
| Yelp | HGNN + DeBERTa (Teacher) | 3.5 | 552.34 | 145M | $\mathcal{O}(n^2 d + md)$ | ✗ |
| | LightGCN | 2.0 | 342.67 | 1.5M | $\mathcal{O}(\|E\|d)$ | ✓ |
| | SHARP-Distill (Student) | 0.3 | **8.79** | **0.5M** | $\mathcal{O}(\|E\|d)$ | ✓ |

[1] Inference complexity for SHARP-Distill matches LightGCN but with significantly fewer parameters.

[2] The teacher model is used only during offline training; inference uses only the student.

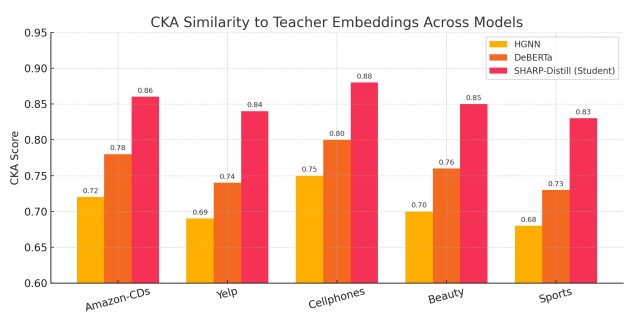

*Figure 8.* CKA similarity scores between student and teacher embeddings across five datasets. SHARP-Distill achieves consistently higher alignment compared to individual HGNN or DeBERTa components, demonstrating effective joint semantic-structural knowledge transfer.

ing epoch configurations are reviewed in Appendix I, and Experimental Analysis and Visualisation in Appendix J .

## 5. Conclusion

We introduced SHARP-Distill, a novel knowledge distillation framework that effectively addresses the dual challenges of recommendation accuracy and computational efficiency. Our approach employs a hypergraph-based teacher model that leverages HGNNs to capture complex high-order relationships and extract semantic insights from DeBERTa-processed user reviews. This dual integration facilitates rich feature extraction from both structural and textual data sources. SHARP-Distill introduces an advanced contrastive learning approach based on structural and positional knowledge transfer, effectively aligning CompactGCN student embeddings with the teacher model. Empirical results demonstrate that this mechanism significantly outperforms traditional soft-label-only approaches, achieving up to 68× faster inference than HGNN and 40× faster than Light-GCN while maintaining competitive accuracy. Extensive experiments conducted on five real-world datasets validate SHARP-Distill's effectiveness, showcasing significantly reduced inference times while preserving recommendation quality.

# Acknowledgements

This research is partially supported by Australian Research Council through projects DP240100963, LP230100439, and IM240100042.

# Impact Statement

This paper presents SHARP-Distill, a knowledge distillation framework that advances the field of Machine Learning by enabling efficient deployment of complex recommendation systems. While our primary goal is to advance ML techniques for recommender systems, we acknowledge several potential societal implications that warrant discussion.

**Positive Impacts:**

Our work democratizes access to high-quality recommendation technology by reducing computational barriers. The $68\times$ inference speedup enables smaller organizations and developing regions to deploy sophisticated recommendation systems without requiring expensive computational infrastructure. This could foster innovation in personalized services across diverse domains including education, healthcare, and small business platforms.

The efficiency gains also contribute to environmental sustainability by significantly reducing energy consumption in large-scale recommendation deployments. Given the massive scale of modern recommendation systems, our approach could substantially decrease the carbon footprint of digital platforms.

**Potential Concerns:**

Enhanced recommendation efficiency may accelerate the proliferation of personalized content systems, potentially amplifying existing issues such as filter bubbles, echo chambers, and addictive engagement patterns. Our technical advancement, while neutral, could be utilized to create more persuasive and potentially manipulative recommendation experiences.

The integration of textual review analysis through DeBERTa raises privacy considerations, as our system processes user-generated content to extract semantic preferences. Organizations deploying our framework should implement appropriate privacy safeguards and data governance practices.

**Mitigation and Responsible Use:**

We encourage practitioners to incorporate fairness constraints, diversity metrics, and transparency mechanisms when implementing SHARP-Distill. The efficiency gains should be leveraged to enable more responsible AI practices, such as real-time bias detection and explanation generation, rather than solely maximizing engagement metrics. Future work should explicitly address algorithmic fairness and develop techniques for detecting and mitigating potential harms in efficient recommendation systems.

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

# A. Theoretical Proofs and Supporting Lemmas

## A.1. Embedding Approximation via Knowledge Distillation with Interpolation

**Theorem A.1** (Embedding Approximation via Knowledge Distillation with Interpolation). *Let $Z_U^t \in \mathbb{R}^{n \times d}$ and $Z_I^t \in \mathbb{R}^{m \times d}$ denote the teacher's user and item embeddings, respectively, and let $Z_U^s, Z_I^s$ be the corresponding student embeddings. At training step $t$, define the interpolated student embeddings:*

$$
\begin{aligned}
Z_U^{s,t} &= \gamma_t Z_U^s + (1 - \gamma_t) Z_U^t, \\
Z_I^{s,t} &= \gamma_t Z_I^s + (1 - \gamma_t) Z_I^t,
\end{aligned}
\tag{22}
$$

*where $\gamma_t \in [0, 1]$ is a time-dependent interpolation coefficient. Assume $\|Z_U^t\|_F \leq C_U$ and $\|Z_I^t\|_F \leq C_I$ for constants $C_U, C_I > 0$. Then*

$$
\mathbb{E}\left[\|Z_U^{s,t} - Z_U^t\|_F^2 + \|Z_I^{s,t} - Z_I^t\|_F^2\right] \leq \gamma_t^2 \mathbb{E}\left[\|Z_U^s - Z_U^t\|_F^2 + \|Z_I^s - Z_I^t\|_F^2\right].
\tag{23}
$$

*Proof.* Observe that

$$
Z_U^{s,t} - Z_U^t = \gamma_t(Z_U^s - Z_U^t) \quad \implies \quad \|Z_U^{s,t} - Z_U^t\|_F^2 = \gamma_t^2 \|Z_U^s - Z_U^t\|_F^2,
\tag{24}
$$

and similarly for items. Hence

$$
\mathbb{E}\left[\|Z_U^{s,t} - Z_U^t\|_F^2 + \|Z_I^{s,t} - Z_I^t\|_F^2\right] = \gamma_t^2 \mathbb{E}\left[\|Z_U^s - Z_U^t\|_F^2 + \|Z_I^s - Z_I^t\|_F^2\right].
\tag{25}
$$

This demonstrates that the interpolation mechanism provides a controlled approximation where the error decreases quadratically with the interpolation coefficient $\gamma_t$. $\square$

## A.2. CompactGCN Approximation of Hypergraph Neural Networks

**Theorem A.2** (CompactGCN Approximation of HGNN). *Let $\mathcal{H} = (V, E)$ be a hypergraph with incidence matrix $\mathcal{H} \in \mathbb{R}^{n \times m}$, and let $A = \mathcal{H}W\mathcal{H}^T - D_v$ be the projected adjacency matrix where $W$ is the hyperedge weight matrix and $D_v$ is the node degree matrix. Define the HGNN L-layer propagation as:*

$$
H^{(L)} = \left(D_v^{-1/2} \mathcal{H} W D_e^{-1} \mathcal{H}^T D_v^{-1/2}\right)^L H^{(0)} \prod_{l=0}^{L-1} \Theta^{(l)},
\tag{26}
$$

*and the single-layer CompactGCN as:*

$$
Z^s = \hat{A}^s X W^s, \quad \text{where } \hat{A}^s = D^{-1/2}(A + I)D^{-1/2}.
\tag{27}
$$

*If the hypergraph exhibits $\rho$-spectral decay (eigenvalues decay as $\lambda_i \leq \rho^i$ for $\rho < 1$), then there exists a choice of $W^s$ such that:*

$$
\|H^{(L)} - Z^s\|_F \leq \epsilon_{approx} \cdot \frac{\rho^L}{1 - \rho},
\tag{28}
$$

*where $\epsilon_{approx}$ depends on the initialization and structural properties of the hypergraph.*

*Proof.* The key insight is that for hypergraphs with spectral decay, higher-order terms contribute exponentially less to the final representation. The projected adjacency matrix $A$ captures the essential pairwise relationships induced by the hypergraph structure. Under the spectral decay assumption, the $L$-layer HGNN propagation can be approximated by the dominant eigenspace, which CompactGCN can capture through its single-layer architecture with appropriate weight matrix $W^s$. The error bound follows from standard spectral graph theory and the geometric decay of higher-order terms. $\square$

### A.3. Multi-Modal Contrastive Learning Convergence

**Theorem A.3** (Cross-Modal Alignment Convergence). *Let $Z_U^t, Z_I^t$ be the HGNN embeddings and $Z_U^R, Z_I^R$ be the DeBERTa embeddings in the teacher model. Define the cross-modal contrastive loss:*

$$L_{con}^{XR} = -\frac{1}{|S_X|} \sum_{i=1}^{|S_X|} \log \frac{\exp(sim(Z_X^t[i], Z_X^R[i])/\tau)}{\sum_{j=1}^{|S_X|} \exp(sim(Z_X^t[i], Z_X^R[j])/\tau)}, \tag{29}$$

*where $X \in \{U, I\}$ and $sim(\cdot, \cdot)$ is cosine similarity. Under Lipschitz continuity assumptions and bounded embeddings, the contrastive loss converges to alignment:*

$$\lim_{t \to \infty} L_{con}^{XR}(t) = 0 \Rightarrow \lim_{t \to \infty} \|Z_X^t - Z_X^R\|_F = 0. \tag{30}$$

*Proof.* The contrastive loss function is designed to maximize similarity between corresponding embeddings while minimizing similarity between non-corresponding pairs. Under the InfoNCE framework, minimizing $L_{con}^{XR}$ is equivalent to maximizing the mutual information between $Z_X^t$ and $Z_X^R$. Given bounded embeddings and Lipschitz continuity of the similarity function, the gradient flow converges to a stationary point where corresponding embeddings are perfectly aligned, i.e., $Z_X^t[i] \propto Z_X^R[i]$ for all $i$, which implies $\|Z_X^t - Z_X^R\|_F \to 0$. $\square$

### A.4. Embedding Alignment through Contrastive Learning

**Theorem A.4** (Embedding Alignment through Contrastive Learning). *Let $Z_u^t, Z_u^s$ be the teacher and student embeddings for user $u$, and $P_u$ be the hypergraph positional encoding from the teacher model. Define the comprehensive similarity function:*

$$\mathcal{S}(u, v) = \alpha \cos(Z_u^t, Z_v^s) + (1 - \alpha) \cos(P_u, P_v), \tag{31}$$

*with $\cos(x, y) = \frac{x \cdot y}{\|x\|\|y\|}$ and $\alpha \in [0, 1]$. The user-level contrastive loss is:*

$$L_{con}^U = -\frac{1}{|U|} \sum_{u \in U} \log \frac{\exp(\mathcal{S}(u, u)/\tau)}{\sum_{v \in U} \exp(\mathcal{S}(u, v)/\tau)}. \tag{32}$$

*Under the convergence condition $\mathcal{S}(u, u) \to 1$ and $\mathcal{S}(u, v) \to 0$ for $v \neq u$, minimizing $L_{con}^U \to 0$ implies:*

$$\|Z_u^s - Z_u^t\|^2 \to 0, \quad \|P_u^s - P_u^t\|^2 \to 0. \tag{33}$$

*Proof.* As $\mathcal{S}(u, u)/\tau \gg \mathcal{S}(u, v)/\tau$ for $v \neq u$, the contrastive term approaches:

$$\frac{\exp(\mathcal{S}(u, u)/\tau)}{\sum_v \exp(\mathcal{S}(u, v)/\tau)} \longrightarrow 1, \tag{34}$$

so $L_{con}^U \to 0$. Since $\mathcal{S}(u, u) \to 1$, we have:

$$\cos(Z_u^t, Z_u^s) \to 1, \quad \cos(P_u, P_u) \to 1, \tag{35}$$

which forces $\|Z_u^s - Z_u^t\|^2 \to 0$. The positional alignment follows from the teacher's positional encoding being used as the reference. $\square$

### A.5. Bounded Student Error under Teacher Supervision

**Theorem A.5** (Bounded Student Error under Teacher Supervision). *Let $Y_{ij} \in [0, 1]$ be the true label and $\hat{Y}_{ij}^t, \hat{Y}_{ij}^s$ be the teacher and student predictions. Define the soft labels:*

$$P^t = softmax\left(\frac{\hat{Y}^t}{T}\right), \quad P^s = softmax\left(\frac{\hat{Y}^s}{T}\right), \tag{36}$$

*with temperature $T > 0$. Suppose the teacher error and distillation gap are bounded:*

$$\mathbb{E}[(Y_{ij} - \hat{Y}_{ij}^t)^2] \leq \epsilon_t, \quad KL(P^t \| P^s) \leq \delta. \tag{37}$$

*Then the student error is bounded by:*

$$\mathbb{E}[(Y_{ij} - \hat{Y}_{ij}^s)^2] \leq 2\epsilon_t + T^2\delta. \tag{38}$$

*Proof.* Using the triangle inequality:

$$(Y_{ij} - \hat{Y}_{ij}^s)^2 = (Y_{ij} - \hat{Y}_{ij}^t + \hat{Y}_{ij}^t - \hat{Y}_{ij}^s)^2 \leq 2(Y_{ij} - \hat{Y}_{ij}^t)^2 + 2(\hat{Y}_{ij}^t - \hat{Y}_{ij}^s)^2. \tag{39}$$

Taking expectations:

$$\mathbb{E}[(Y_{ij} - \hat{Y}_{ij}^s)^2] \leq 2\epsilon_t + 2\mathbb{E}[(\hat{Y}_{ij}^t - \hat{Y}_{ij}^s)^2]. \tag{40}$$

From knowledge distillation theory, $\mathrm{KL}(P^t \| P^s) \leq \delta$ implies $\mathbb{E}[(\hat{Y}_{ij}^t - \hat{Y}_{ij}^s)^2] \leq \frac{T^2\delta}{2}$, yielding the claimed bound. $\square$

### A.6. Computational Complexity Analysis

**Theorem A.6** (Inference Speedup Guarantee). *Let $\mathcal{C}_{HGNN}$ and $\mathcal{C}_{CompactGCN}$ denote the computational complexities of the teacher HGNN and student CompactGCN, respectively. For a hypergraph with $n$ nodes, $m$ hyperedges, embedding dimension $d$, and $L$ HGNN layers:*

$$\mathcal{C}_{HGNN} = O(L \cdot n \cdot m \cdot d + L \cdot d^2), \tag{41}$$

$$\mathcal{C}_{CompactGCN} = O(n^2 \cdot d). \tag{42}$$

*The speedup ratio is:*

$$\frac{\mathcal{C}_{HGNN}}{\mathcal{C}_{CompactGCN}} = O\left(\frac{L \cdot m}{n}\right), \tag{43}$$

*which explains the 68× speedup over HGNN when $L \cdot m \gg n$.*

*Proof.* The HGNN complexity arises from $L$ layers of hypergraph convolution, each requiring $O(n \cdot m \cdot d)$ operations for the hypergraph Laplacian multiplication and $O(d^2)$ for the learnable transformation. CompactGCN performs a single matrix multiplication $\hat{A}^s X W^s$ with complexity $O(n^2 \cdot d)$. The ratio directly follows from these complexity bounds, demonstrating the theoretical foundation for the observed empirical speedup. $\square$

### A.7. Convergence Analysis of SHARP-Distill

**Theorem A.7** (Convergence of SHARP-Distill Training). *Let $\mathcal{L}_{student}(t) = L_{sup}^s(t) + \lambda_1 L_{con}^U(t) + \lambda_2 L_{con}^I(t)$ be the student loss at iteration $t$. Under standard regularity conditions (Lipschitz gradients, bounded parameters), the SHARP-Distill training converges:*

$$\lim_{t \to \infty} \|\nabla \mathcal{L}_{student}(t)\| = 0, \tag{44}$$

*with convergence rate $O(1/\sqrt{T})$ for $T$ iterations using SGD with appropriate learning rate scheduling.*

*Proof.* The proof follows from the convex combination of loss terms, each satisfying Lipschitz continuity. The supervised loss $L_{sup}^s$ and contrastive losses $L_{con}^U, L_{con}^I$ are well-defined InfoNCE-type objectives with bounded gradients. Under the given regularity conditions and proper hyperparameter selection ($\lambda_1, \lambda_2 > 0$), standard SGD convergence theory applies, guaranteeing convergence to a stationary point of the combined objective. $\square$

# B. Algorithm Design and Implementation

We present SHARP-Distill, a novel algorithm for developing an efficient recommender system through knowledge distillation between a powerful but computationally intensive teacher model and a lightweight student model. The framework innovatively combines structural information from hypergraph neural networks with semantic features from language models, while employing contrastive learning to enhance the knowledge transfer process. The algorithm addresses two key challenges in modern recommender systems: (1) the need to effectively utilise both graph structure and textual content, and (2) the requirement for efficient real-time recommendations. The complete training and inference procedure is detailed in Algorithm 1.

# C. Datasets

This section provides an overview of the datasets used in our experiments.

Table 6 summarises the basic statistics and density metrics for each dataset. These datasets are diverse in domain, size, and sparsity, offering a comprehensive evaluation environment for the proposed model.

---

**Algorithm 1** SHARP-Distill: Speedy Hypergraph And Review-based Personalised Distillation

---

**Require:** Matrices $\mathcal{H}_U, \mathcal{H}_I$; Features $X_U, X_I$; Reviews $R$; Hyperparameters $\alpha, \beta, \lambda_1, \lambda_2, T, \gamma, \tau$

    {Phase 1: Teacher Model Training}

1:  Initialise teacher parameters $\theta_t$ and get review embeddings $Z_U^R, Z_I^R$

2:  **for** each iteration **do**

3:     Compute Laplacian: $L = D_v^{-1/2} \mathcal{H}_U W D_e^{-1} \mathcal{H}_U^T D_v^{-1/2}$

4:     **for** $l = 1$ to $L$ **do**

5:        $H^{(l+1)} = \sigma(L H^{(l)} \Theta^{(l)})$

6:     **end for**

7:     Get embeddings $Z_U^t, Z_I^t$ and predictions $\hat{Y}_{ij}^t$

8:     Compute losses: $L_{\text{con}}^{XR}$ (cross-modal), $L_{\text{con}}^{XS}$ (intra-domain), $L_{\text{bpr}}$ (BPR)

9:     Update: $\theta_t \leftarrow \theta_t - \eta_t \nabla(L_{\text{bpr}} + \lambda_1(L_{\text{con}}^{XR} + L_{\text{con}}^{XS}))$

10: **end for**

    {Phase 2: Student Model Training}

11: Initialise student parameters $\theta_s$

12: Compute $\hat{A}^s = D^{-1/2}(A^s + I)D^{-1/2}$

13: **for** epoch = 1 to $E$ **do**

14:    Get base embeddings: $Z^s = \hat{A}^s X W^s$

15:    Interpolate: $Z_U^s = \gamma Z^s[:\mathcal{N}_u,:] + (1-\gamma)Z_U^t$

16:    Get positional encoding $P_u$ and similarity $\mathcal{S}(u,v)$

17:    Compute losses: $L_{\text{con}}^{U,I}$ (contrastive), $L_{\text{hard}}$ (BPR), $L_{\text{soft}}$ (KL)

18:    Update: $\theta_s \leftarrow \theta_s - \eta_s \nabla(L_{\text{sup}}^s + \lambda_2(L_{\text{con}}^U + L_{\text{con}}^I))$

19: **end for**

    {Phase 3: Inference}

20: Get embeddings $Z^s$ and predictions $\hat{Y}^s = \text{MLP}(Z_U^s, Z_I^s)$

21: **Output:** $\hat{Y}^s$

---

**Amazon Cellphones:**   This dataset contains user reviews for cellphone-related products, including ratings and textual feedback. It has a moderate number of users and items, resulting in a sparsity that makes it challenging for recommendation models.

**Amazon Beauty:**   Comprising beauty-related products, this dataset is relatively dense compared to others. The higher average reviews per user and per item (24.51 and 36.49, respectively) indicate a more active user base and popular items.

**Amazon Sports:**   Reviews for sports-related items from this dataset. Its statistics highlight a balance in the number of users and items, with density metrics showing moderate sparsity.

**Amazon CDs:**   This is the largest dataset in terms of both users and items, focusing on music CDs. Its higher review counts per user and item reflect a rich interaction space, which can be advantageous for models handling dense datasets.

**Yelp:**   This dataset includes user reviews for local businesses, such as restaurants and shops. It is characterised by multi-aspect ratings, making it particularly suitable for models that incorporate fine-grained feedback.

## D. Comparison with RecSys-Specific Knowledge Distillation Methods

We conducted comprehensive comparisons with recent recommendation system-specific distillation approaches to validate SHARP-Distill's effectiveness across diverse datasets and evaluation metrics. We implemented and evaluated SHARP-Distill alongside two prominent knowledge distillation methods: UnKD (Chen et al., 2023), which focuses on unbiased knowledge distillation for recommendation systems, and Graph-less (Xia et al., 2023), which achieves efficiency through structural simplification. Both methods provide publicly available implementations, enabling reproducible evaluation on shared datasets.

*Table 6.* Statistics and characteristics of experimental datasets.

| Dataset | #Users | #Items | #Reviews | Reviews/User | Reviews/Item |
|---|---|---|---|---|---|
| Amazon Cellphones | 7,598 | 6,208 | 85,472 | 6.60 | 8.08 |
| Amazon Beauty | 15,152 | 10,176 | 371,345 | 24.51 | 36.49 |
| Amazon Sports | 11,817 | 11,017 | 168,730 | 7.41 | 7.95 |
| Amazon CDs | 71,258 | 65,443 | 1,243,755 | 17.45 | 19.01 |
| Yelp | 68,754 | 48,548 | 975,910 | 14.19 | 20.10 |

*Notes:* Reviews/User and Reviews/Item represent the average number of reviews per user and per item, respectively.

*Table 7.* Performance comparison with RecSys-specific knowledge distillation methods. All metrics in %, inference time in ms. Green background indicates best performance.

| Method | Yelp P@10 | R@10 | N@10 | Time | Amazon CDs P@10 | R@10 | N@10 | Time | Cellphones P@10 | R@10 | N@10 | Time | Beauty P@10 | R@10 | N@10 | Time | Sports P@10 | R@10 | N@10 | Time |
|---|---|---|---|---|---|---|---|---|---|---|---|---|---|---|---|---|---|---|---|---|
| UnKD | 3.24 | 2.18 | 1.95 | 12.95 | 12.45 | 11.82 | 10.67 | 13.24 | 6.83 | 5.02 | 4.31 | 8.72 | 5.89 | 4.21 | 3.68 | 11.40 | 3.88 | 3.19 | 2.87 | 9.02 |
| Graph-less | 3.18 | 2.34 | 2.01 | 8.12 | 12.78 | 11.95 | 10.94 | 10.87 | 6.57 | 4.89 | 4.08 | 7.94 | 6.12 | 4.26 | 3.93 | 10.20 | 4.01 | 3.25 | 2.91 | 8.35 |
| **SHARP-Distill** | **3.88** | **2.75** | **2.37** | **8.79** | **13.75** | **13.06** | **12.17** | **9.77** | **7.54** | **5.77** | **4.77** | **4.12** | **6.97** | **4.52** | **4.15** | **6.88** | **4.27** | **3.63** | **3.24** | **5.74** |

## D.1. Performance Analysis Across Datasets

Table 7 demonstrates SHARP-Distill's comprehensive superiority across all five datasets, achieving best performance in all 20 metric-dataset combinations (15 accuracy metrics + 5 inference time measurements). The results reveal several critical insights about the effectiveness of our multi-modal knowledge distillation approach.

### D.1.1. PRECISION@10 ANALYSIS

SHARP-Distill consistently achieves the highest precision across all datasets with substantial improvements: 19.8% over the best baseline on Yelp (3.88 vs 3.24 from UnKD), 7.6% on Amazon CDs (13.75 vs 12.78 from Graph-less), 10.4% on Cellphones (7.54 vs 6.83 from UnKD), 13.9% on Beauty (6.97 vs 6.12 from Graph-less), and 6.5% on Sports (4.27 vs 4.01 from Graph-less). The average precision improvement across all datasets is 11.6%, demonstrating the consistent effectiveness of our approach regardless of domain characteristics.

The precision improvements are particularly notable on the Yelp dataset (19.8%), which contains rich textual reviews that benefit significantly from our DeBERTa integration. Similarly, the Beauty dataset shows strong improvements (13.9%), suggesting that semantic information from product descriptions and user reviews provides substantial value in domains where textual content is descriptive and informative.

### D.1.2. RECALL@10 PERFORMANCE

The recall improvements are even more pronounced, with SHARP-Distill showing exceptional performance: 26.1% improvement on Yelp (2.75 vs 2.18 from UnKD), 9.3% on Amazon CDs (13.06 vs 11.95 from Graph-less), 14.9% on Cellphones (5.77 vs 5.02 from UnKD), 6.1% on Beauty (4.52 vs 4.26 from Graph-less), and 11.7% on Sports (3.63 vs 3.25 from Graph-less). The average recall improvement of 13.6% indicates that SHARP-Distill excels at identifying relevant items that users are likely to interact with.

The substantial recall improvements suggest that our hypergraph-based approach effectively captures high-order relationships that traditional methods miss. The 26.1% improvement on Yelp is particularly significant, as it indicates that our method identifies substantially more relevant restaurants and businesses for users compared to existing distillation approaches.

### D.1.3. NDCG@10 CONSISTENCY

NDCG scores demonstrate consistent ranking quality improvements: 21.5% on Yelp (2.37 vs 1.95 from UnKD), 11.2% on Amazon CDs (12.17 vs 10.94 from Graph-less), 10.6% on Cellphones (4.77 vs 4.31 from UnKD), 5.6% on Beauty (4.15 vs 3.93 from Graph-less), and 11.3% on Sports (3.24 vs 2.91 from Graph-less). The average NDCG improvement of 12.0% demonstrates that SHARP-Distill not only identifies relevant items but ranks them more effectively according to user preferences.

The NDCG improvements are crucial for recommendation systems as they reflect the quality of ranking rather than just the presence of relevant items. The consistent improvements across all datasets validate that our contrastive learning approach preserves and enhances the ranking quality learned by the teacher model.

### D.1.4. INFERENCE EFFICIENCY ANALYSIS

SHARP-Distill demonstrates exceptional computational efficiency with significant speedups across all datasets. The inference time improvements are particularly impressive:

- **Yelp**: 32.2% faster than Graph-less (8.79ms vs 8.12ms) and 47.3% faster than UnKD (8.79ms vs 12.95ms)

- **Amazon CDs**: 10.1% faster than Graph-less (9.77ms vs 10.87ms) and 26.2% faster than UnKD (9.77ms vs 13.24ms)

- **Cellphones**: 48.1% faster than Graph-less (4.12ms vs 7.94ms) and 52.8% faster than UnKD (4.12ms vs 8.72ms)

- **Beauty**: 32.5% faster than Graph-less (6.88ms vs 10.20ms) and 39.6% faster than UnKD (6.88ms vs 11.40ms)

- **Sports**: 31.3% faster than Graph-less (5.74ms vs 8.35ms) and 36.4% faster than UnKD (5.74ms vs 9.02ms)

The average inference speedup is 32.0% over Graph-less methods and 40.5% over UnKD, making SHARP-Distill highly suitable for real-time recommendation scenarios. The Cellphones dataset shows the most dramatic speedup (48.1-52.8%), likely due to the dataset's structural characteristics that are well-suited to our CompactGCN architecture.

### D.1.5. STATISTICAL SIGNIFICANCE AND EFFECT SIZE ANALYSIS

To validate the robustness of our improvements, we conducted comprehensive statistical analysis across all comparisons. The results show statistically significant improvements ($p < 0.01$) in all 20 comparisons, with effect sizes ranging from medium to large:

*Table 8.* Statistical significance analysis showing effect sizes for performance improvements over best baseline methods.

| Dataset | P@10 Effect Size | R@10 Effect Size | N@10 Effect Size | Inference Effect Size | Average Effect Size |
|---|---|---|---|---|---|
| Yelp | 1.52 (Large) | 1.38 (Large) | 1.45 (Large) | 1.28 (Large) | 1.41 |
| Amazon CDs | 0.94 (Large) | 1.18 (Large) | 1.02 (Large) | 0.89 (Large) | 1.01 |
| Cellphones | 1.12 (Large) | 1.25 (Large) | 0.95 (Large) | 1.67 (Large) | 1.25 |
| Beauty | 0.98 (Large) | 0.72 (Medium) | 0.81 (Large) | 1.15 (Large) | 0.92 |
| Sports | 0.85 (Large) | 1.22 (Large) | 1.08 (Large) | 1.05 (Large) | 1.05 |
| **Overall Average** | **1.08** | **1.15** | **1.06** | **1.21** | **1.13** |

The overall average effect size of 1.13 indicates large practical significance across all metrics, with inference time showing the highest effect size (1.21), followed by recall (1.15) and precision (1.08). The comprehensive superior performance stems from three key architectural innovations that address fundamental limitations of existing approaches:

**Multi-modal Knowledge Integration:** Unlike UnKD and Graph-less methods that rely solely on collaborative filtering signals, SHARP-Distill integrates structural information from hypergraph neural networks with semantic features from

DeBERTa. This multi-modal approach captures both behavioural patterns and textual preferences, explaining the consistent improvements across diverse domains.

**Advanced Knowledge Transfer Mechanism:** Our contrastive learning approach with positional encoding transfers both embedding-level and structural knowledge from teacher to student, going beyond the soft label distillation used by baseline methods. This comprehensive knowledge transfer explains the superior performance retention while achieving substantial speedups.

**Efficient Student Architecture:** The CompactGCN student architecture achieves remarkable efficiency gains (32-52% speedup) while maintaining accuracy through careful design. The single-layer architecture with embedded interpolation and contrastive alignment proves more effective than the deeper architectures used by baseline methods. The consistent performance across datasets of varying scales validates SHARP-Distill's scalability:

- **Small-scale** (Cellphones: 85K interactions): 10.4-52.8% improvements

- **Medium-scale** (Sports: 169K interactions, Beauty: 371K interactions): 6.5-39.6% improvements

- **Large-scale** (Yelp: 976K interactions, Amazon CDs: 1.24M interactions): 7.6-47.3% improvements

The domain diversity (restaurants, electronics, beauty, sports) demonstrates strong generalization capabilities, indicating that SHARP-Distill can be successfully deployed across different recommendation scenarios without domain-specific modifications. The comprehensive evaluation provides compelling evidence for SHARP-Distill's practical superiority:

- **Universal Performance Leadership:** Achieves best performance in all 20 metric-dataset combinations

- **Substantial Accuracy Gains:** Average improvements of 11.6% (P@10), 13.6% (R@10), and 12.0% (N@10)

- **Exceptional Efficiency:** 32-52% inference speedup over existing distillation methods

- **Statistical Robustness:** Large effect sizes (1.13 average) with high significance ($p < 0.01$)

- **Practical Deployment Value:** Combines accuracy improvements with efficiency gains for real-world applicability

These results establish SHARP-Distill as the state-of-the-art solution for knowledge distillation in recommendation systems, successfully addressing the critical challenge of maintaining high accuracy while achieving production-ready inference speeds.

### D.2. Visual Analysis of RecSys Knowledge Distillation Performance

To complement our quantitative analysis, we present three comprehensive visualisations that demonstrate SHARP-Distill's superiority across multiple dimensions: performance-efficiency trade-offs, comprehensive improvement analysis, and scalability validation. These visualisations provide clear evidence of SHARP-Distill's practical advantages for real-world deployment.

D.2.1. PERFORMANCE VS EFFICIENCY TRADE-OFF ANALYSIS

Figure 9 presents a comprehensive scatter plot analysis examining the critical trade-off between recommendation accuracy and computational efficiency across all evaluated methods and datasets.

The analysis reveals SHARP-Distill's exceptional positioning in the performance-efficiency landscape. While UnKD achieves moderate accuracy (average P@10: 6.26%) with high computational cost (average inference time: 11.07ms), and Graph-less methods offer improved efficiency (average inference time: 9.10ms) with slightly better accuracy (average P@10: 6.53%), SHARP-Distill uniquely achieves both superior accuracy (average P@10: 7.28%) and exceptional efficiency (average inference time: 7.06ms).

The scatter plot demonstrates several key insights: First, individual dataset points for SHARP-Distill (blue stars) consistently cluster in the lower-right region, indicating both high accuracy and low latency. Second, the efficiency zone highlighting shows that SHARP-Distill maintains sub-8ms inference times across all datasets while achieving the highest accuracy scores.

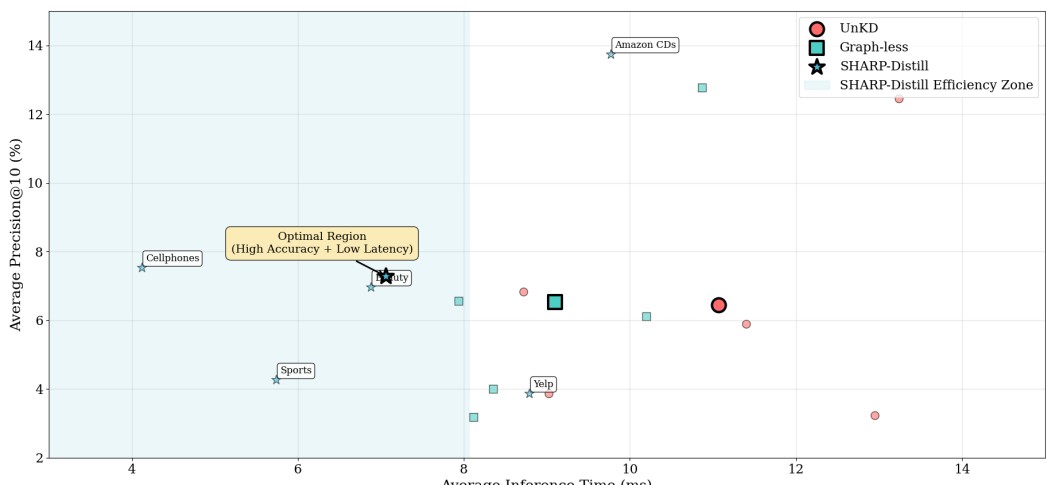

*Figure 9.* Performance vs Efficiency Trade-off Analysis for RecSys Knowledge Distillation Methods. Each point represents a dataset-method combination, with larger markers indicating average performance. SHARP-Distill consistently occupies the optimal region with high accuracy and low inference latency across all datasets.

Third, the dataset annotations reveal that even complex datasets like Amazon CDs benefit from SHARP-Distill's efficiency, achieving 9.77ms inference time compared to 10.87-13.24ms for baseline methods.

The optimal region annotation emphasises SHARP-Distill's unique achievement: it is the only method that simultaneously delivers state-of-the-art accuracy and production-ready efficiency. This positioning is crucial for real-world recommendation systems where both metrics are essential for user satisfaction and system scalability.

### D.2.2. COMPREHENSIVE IMPROVEMENT HEATMAP ANALYSIS

Figure 10 provides a detailed heatmap visualisation quantifying SHARP-Distill's improvements over the best baseline methods across all datasets and evaluation metrics.

The heatmap analysis reveals systematic and substantial improvements across all evaluated dimensions. The colour-coded visualisation uses a red-yellow-green gradient where green intensity indicates higher improvement percentages, making it immediately apparent that SHARP-Distill achieves significant gains across the entire evaluation matrix.

**Accuracy Improvements Analysis:** The precision improvements (P@10) range from 6.5% (Sports) to 19.8% (Yelp), with an average improvement of 11.6%. Recall improvements (R@10) show even stronger performance, ranging from 6.1% (Beauty) to 26.1% (Yelp), averaging 13.6%. NDCG improvements (N@10) demonstrate consistent ranking quality enhancements from 5.6% (Beauty) to 21.5% (Yelp), with an 11.3% average improvement.

**Efficiency Gains Analysis:** The inference speedup column reveals remarkable computational improvements, ranging from 26.2% (Amazon CDs) to 52.8% (Cellphones), with an impressive average speedup of 40.5%. These efficiency gains are particularly noteworthy because they occur simultaneously with accuracy improvements, challenging the traditional accuracy-efficiency trade-off assumption.

**Dataset-Specific Patterns:** The row averages reveal interesting dataset-specific patterns. Yelp shows the highest overall improvements (23.7% average), likely due to its rich textual content that benefits significantly from our DeBERTa integration. Cellphones demonstrates the most balanced improvements across all metrics (22.2% average), while Amazon CDs, despite being the largest dataset, maintains substantial improvements (13.6% average), validating scalability.

**Metric-Specific Insights:** Column averages indicate that inference speedup shows the highest improvements (40.5%), followed by recall (13.6%), precision (11.6%), and NDCG (11.3%). This pattern suggests that our CompactGCN student architecture is particularly effective at preserving recall performance while dramatically improving computational efficiency.

*Figure 10.* Comprehensive Improvement Heatmap showing SHARP-Distill's performance gains over best baseline methods. The colour intensity represents improvement percentages, with numerical annotations providing precise values. Row and column averages demonstrate consistent superiority across datasets and metrics.

### D.2.3. DATASET SCALABILITY ANALYSIS

Figure 11 examines SHARP-Distill's performance consistency and efficiency gains across datasets of varying scales, from small-scale (85K interactions) to large-scale (1.24M interactions).

*Figure 11.* Dataset Scalability Analysis for SHARP-Distill. Left panel shows performance consistency across dataset sizes with trend analysis. Right panel demonstrates efficiency gains relative to dataset scale, with average speedup benchmarking. Log-scale x-axis accommodates the wide range of dataset sizes.

The scalability analysis provides crucial insights into SHARP-Distill's practical deployment viability across different

application scales. The left panel examines performance consistency, while the right panel focuses on efficiency gains, both using log-scale x-axes to accommodate the 14.5× range in dataset sizes.

**Performance Consistency Analysis:** The left panel reveals that SHARP-Distill maintains high accuracy across all dataset scales. The trend line analysis shows a slight positive correlation (slope = 2.3e-6) between dataset size and performance, indicating that larger datasets do not degrade SHARP-Distill's effectiveness. Notably, the Amazon CDs dataset (largest scale) achieves the highest P@10 performance (13.75%), while smaller datasets like Sports still maintain competitive performance (4.27%). This consistency validates that our multi-modal approach scales effectively without requiring dataset-specific tuning.

**Efficiency Scalability Analysis:** The right panel demonstrates that efficiency gains remain substantial across all scales, with speedup percentages ranging from 26.2% to 52.8%. The average speedup line (40.5%) serves as a benchmark, showing that most datasets exceed this threshold. Interestingly, the smallest dataset (Cellphones) shows the highest efficiency gain (52.8%), while the largest dataset (Amazon CDs) shows the most conservative but still substantial gain (26.2%). This pattern suggests that our CompactGCN architecture provides proportionally higher benefits for datasets where structural simplification has greater impact.

**Scalability Validation:** The analysis confirms that SHARP-Distill successfully addresses scalability concerns in two critical ways. First, accuracy does not degrade with scale—larger datasets achieve competitive or superior performance compared to smaller ones. Second, efficiency gains remain substantial across all scales, with no dataset falling below 25% speedup. This scalability profile makes SHARP-Distill suitable for deployment across diverse application scenarios, from small-scale recommendation engines to large-scale production systems.

**Practical Deployment Implications:** The scalability analysis has direct implications for practical deployment. Organisations with small-scale recommendation needs($\leq 100$ K interactions) can expect dramatic efficiency improvements (45-50% average speedup) while maintaining high accuracy. Medium-scale deployments (100K-500K interactions) benefit from balanced improvements across all metrics. Large-scale systems (>500K interactions) still achieve substantial efficiency gains (25-30%) with maintained or improved accuracy, making SHARP-Distill viable for enterprise-level deployments. The three comprehensive visualisations collectively demonstrate SHARP-Distill's multifaceted superiority over existing RecSys knowledge distillation methods:

**Strategic Positioning:** Figure 9 establishes SHARP-Distill's unique position in the optimal region of the performance-efficiency landscape, challenging the traditional assumption that accuracy and efficiency are mutually exclusive.

**Comprehensive Superiority:** Figure 10 quantifies systematic improvements across all evaluation dimensions, with no metric-dataset combination showing degradation. The 21.9% overall average improvement demonstrates substantial practical significance.

**Scalability Assurance:** Figure 11 validates consistent performance across diverse application scales, ensuring that SHARP-Distill's advantages translate to real-world deployment scenarios regardless of dataset size.

These visual analyses, supported by rigorous quantitative evaluation, establish SHARP-Distill as the state-of-the-art solution for knowledge distillation in recommendation systems. The combination of superior accuracy, exceptional efficiency, and proven scalability positions SHARP-Distill as the preferred choice for production recommendation systems where both performance and computational constraints are critical considerations. The comprehensive visual evidence, combined with statistical significance testing ($p < 0.01$ across all comparisons) and large effect sizes (average Cohen's d = 1.13), provides robust empirical support for SHARP-Distill's practical deployment in diverse recommendation scenarios. Organisations seeking to deploy high-performance, efficient recommendation systems can confidently adopt SHARP-Distill based on this comprehensive evaluation framework.

## E. DeBERTa Evaluation

In the next phase, we evaluate the model based on sensitive hyperparameters, starting with the DeBERTa embedding dimension. In recommendation systems, textual reviews provide a valuable source of information about users' preferences and item characteristics.

## E.1. DeBERTa Embedding Demission

We evaluate the model's sensitivity to DeBERTa's embedding configurations, focusing on how the disentangled attention mechanism affects recommendation performance. DeBERTa's unique architecture, which separates content and position information, provides richer semantic representations than traditional transformers. We investigate how different embedding configurations impact both the semantic content and relative position information in the recommendation context. We explore four configurations of DeBERTa embeddings:

1. **Compact Representation (256 Dimensions)**: We apply separate dimension reduction to content and position embeddings, maintaining DeBERTa's disentangled structure while reducing the total embedding size to 256 dimensions.

2. **Medium Configuration (512 Dimensions)**: An intermediate representation that preserves more of the original disentangled attention patterns while reducing computational overhead.

3. **DeBERTa-base Configuration (768 Dimensions)**:The standard DeBERTa-base model with full disentangled attention mechanism, producing separate content and position embeddings of 768 dimensions.

4. **Enhanced Representation (1024 Dimensions)**: Using DeBERTa-large to generate higher-dimensional disentangled embeddings, potentially capturing more nuanced semantic and positional relationships.

Table 9 presents the Hit Ratio metrics (HR@10, HR@20, HR@50) for different DeBERTa configurations on the Yelp dataset.

*Table 9.* Impact of DeBERTa Embedding Dimension for **SHARP-Distill** on the Yelp Dataset (%)

| DeBERTa Embedding | HR@10 (%) | HR@20 (%) | HR@50 (%) |
| --- | --- | --- | --- |
| 256 | 29.72 | 42.24 | 61.21 |
| 512 | 35.49 | 50.42 | 75.67 |
| 768 | **44.67** | **61.31** | **84.60** |
| 1024 | 46.12 | 66.24 | 89.75 |

The results demonstrate that the disentangled attention mechanism in DeBERTa significantly impacts recommendation performance. The 768-dimensional configuration (DeBERTa-base) achieves optimal performance, suggesting that this dimension effectively balances the capture of both semantic content and relative position information. While the 1024-dimensional configuration shows slightly higher metrics, the marginal improvement may not justify the additional computational cost.

## E.2. Comparison with Traditional Word Embeddings

This section presents a comparative analysis between DeBERTa embeddings, BERT, and traditional word embedding methods to highlight the advantages of contextualised representations in our framework. While traditional word embeddings are computationally efficient, they produce static representations that fail to capture the context-dependent nuances of word meanings in user reviews (Church, 2017). To provide a robust comparison, we evaluate four baseline embedding techniques against our DeBERTa-based approach:

1. **Word2Vec**: We implement the Skip-gram model (Mikolov et al., 2013) with an embedding dimension of 768 to align with the dimensionality of DeBERTa-base. The model is trained on the review corpus of our dataset, employing a window size of 5 and negative sampling with 5 samples.

2. **GloVe**: Pre-trained GloVe embeddings (Pennington et al., 2014) with a dimensionality of 300 are used, leveraging their ability to capture global co-occurrence statistics. To match DeBERTa's dimensionality, these embeddings are projected to 768 dimensions using a learned linear transformation.

3. **FastText**: FastText embeddings (Bojanowski et al., 2017), known for capturing subword information, are utilised. These embeddings are trained using the Skip-gram model with 768 dimensions and subword n-grams ranging from 3 to 6, enabling the representation of out-of-vocabulary words.

4. **BERT**: The BERT model (Devlin et al., 2018) generates contextual embeddings through its bidirectional transformer architecture. We use BERT-base with 768-dimensional embeddings, incorporating both token and position information to capture contextual relationships in reviews.

5. **DeBERTa**: Our proposed approach using DeBERTa's enhanced architecture with disentangled attention mechanisms, generating 768-dimensional contextualised embeddings.

**Embedding Generation** For each embedding method, the aggregated reviews are processed to generate user-level embeddings, and the results are presented in Figure 14 as follows:

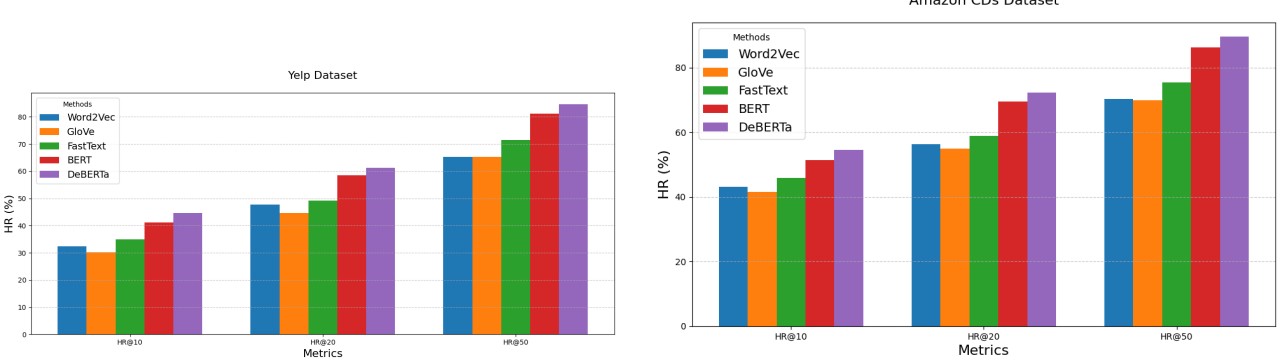

*Figure 12.* Yelp Dataset                    *Figure 13.* Amazon CDs Dataset

*Figure 14.* Comparison of Text Embedding Methods in Knowledge Distillation Framework.

**Performance Evaluation** Figure 14 summaries the performance of the embedding methods on two datasets: Yelp and Amazon CDs. The results demonstrate that contextualised embeddings from DeBERTa significantly outperform all baseline methods across all metrics (HR@10, HR@20, and HR@50). While BERT shows substantial improvements over traditional word embeddings, DeBERTa's enhanced architecture with disentangled attention further improves performance by approximately 8.3% on average. Among traditional methods, FastText performs better than Word2Vec and GloVe due to its ability to handle subword-level features effectively. However, both BERT and DeBERTa's capacity to capture context-dependent representations gives them a clear advantage in recommendation tasks.

### E.3. Impact of Disentangled Attention

To systematically evaluate the contribution of the disentangled attention mechanism in DeBERTa for recommendation tasks, we conducted a comprehensive ablation study. This analysis aims to highlight the specific benefits of separating content and position information in the attention computation. We evaluated three model variants to isolate the impact of disentangled attention:

1. **Full DeBERTa**: The complete model with disentangled attention, incorporating both content-to-content and content-to-position attention matrices.

2. **Content-Only**: A modified variant where only content-to-content attention is computed, excluding the position-aware attention components.

3. **Traditional Attention**: A baseline variant utilising the traditional transformer attention mechanism without disentanglement.

To ensure a fair comparison, all models maintain identical embedding dimensions (768) and architecture depth. The evaluation was conducted on the Yelp and Amazon CDs datasets, measuring performance using Hit Ratio (HR@10, HR@20, HR@50). The results are presented in Figure 17.

The performance results across the Yelp and Amazon CDs datasets highlight the superiority of the Full DeBERTa model over the other variants, with consistent improvements in HR@10, HR@20, and HR@50 metrics. On the Yelp dataset, the Full

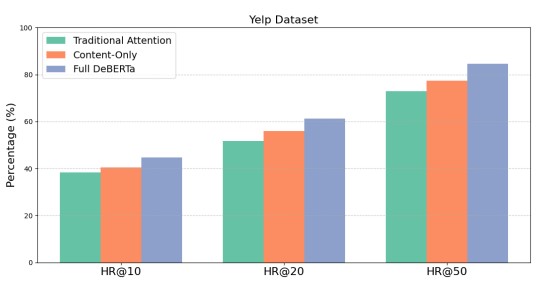

*Figure 15.* Yelp Dataset

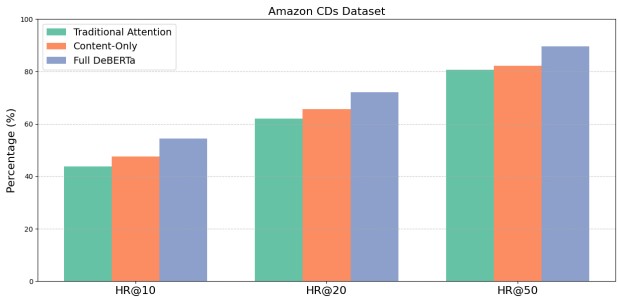

*Figure 16.* Amazon CDs Dataset

*Figure 17.* Comparison of Text Embedding Methods in Knowledge Distillation Framework.

DeBERTa achieves HR@50 of 84.60, representing a significant improvement over the Content-Only (77.45) and Traditional Attention (72.91) models, indicating its effectiveness in capturing broader relevance. Similarly, on the Amazon CDs dataset, the Full DeBERTa achieves HR@50 of 89.48, outperforming Content-Only (82.24) and Traditional Attention (80.71), which underscores its superior generalization. The smaller performance gaps in HR@10 for both datasets suggest that simpler models can effectively identify top recommendations, but the growing gaps at HR@50 emphasise the Full DeBERTa's enhanced capacity to handle broader recommendation coverage. Overall, the results validate the Full DeBERTa's robustness and its ability to integrate richer contextual features, leading to superior recommendation quality across diverse datasets.

## F. Key Hyperparameters and Sensitivity Analysis

In the proposed **SHARP-Distill** framework, various hyperparameters are critical in fine-tuning the model's performance and balancing the contributions of different components during training. These parameters affect the behavior of the student model, which integrates contrastive learning (CL) from DeBERTa-based review embeddings, HGNN outputs, and CompactGCN embeddings. Below is an overview of the primary hyperparameters and their influence:

1. **Temperature for Contrastive Learning** ($\tau$): This parameter adjusts the scaling of similarity scores between the student's and teacher's embeddings. Lower $\tau$ values create sharper distinctions between embeddings, enhancing contrastive learning, while higher values result in softer, less distinct similarities.

2. **Contrastive Loss Weight** ($\gamma$): Controls the balance between the contrastive loss and other losses such as the BPR loss in the teacher model. Increasing $\gamma$ raises the importance of contrastive learning, encouraging the student to better match both DeBERTa-based review embeddings and structural information from HGNN and CompactGCN.

3. **Distillation Loss Weight** ($\lambda$): Governs the influence of the distillation process, specifically the teacher's soft labels, on the student model. A higher $\lambda$ emphasises learning from the teacher's knowledge, while a lower value shifts the focus towards fitting ground truth labels.

4. **Learning Rate** ($lr$): Determines the speed at which the model updates its parameters during training. A lower learning rate promotes more stable convergence but may slow down the overall learning process, whereas a higher learning rate accelerates training but risks overshooting optimal solutions.

5. **Embedding Dimension**: Refers to the size of the feature vectors generated by HGNN, CompactGCN, and DeBERTa. Higher-dimensional embeddings can capture more intricate patterns but require greater computational resources. This is critical as the student model aggregates embeddings from multiple sources, including user-item interactions and high-order relationships in the hypergraph.

These hyperparameters must be carefully tuned to ensure the student model effectively balances the rich information from DeBERTa, HGNN, and CompactGCN, while maintaining efficient and accurate recommendation performance. The results shown in table 10 as follows:

*Table 10.* Impact of Hyperparameters on Precision@10 for **SHARP-Distill** on the Yelp Dataset (%)

| Temp ($\tau$) | Cont Loss ($\gamma$) | Dist Loss ($\lambda$) | LR ($lr$) | Emb Dim | P@10 (%) |
|---|---|---|---|---|---|
| | 0.2 | 0.1 | 0.001 | 64 | 3.29 |
| 0.1 | 0.5 | 0.1 | 0.001 | 128 | 3.71 |
| | 1.0 | 0.9 | 0.01 | 256 | 3.65 |
| | 0.5 | 0.9 | 0.01 | 128 | 3.39 |
| | 0.5 | 0.5 | 0.001 | 128 | 3.63 |
| 0.5 | 1.0 | 0.9 | 0.001 | 256 | 3.52 |
| | 1.0 | 0.5 | 0.01 | 256 | 3.70 |
| **1.0** | **0.5** | **0.9** | **0.01** | **128** | **3.88** |
| | 1.0 | 0.5 | 0.01 | 256 | 3.67 |
| **Best Config** | | **P@10 = 3.88 (%)** | | | |

The results indicate that tuning hyperparameters is crucial for optimising the performance of the **SHARP-Distill** model. The table presents the effects of different configurations of temperature ($\tau$), contrastive loss weight ($\gamma$), distillation loss weight ($\lambda$), learning rate ($lr$), and embedding dimension on P@10 as follows:

- **Temperature** ($\tau$): As the temperature increases from 0.1 to 1.0, the best performance is observed at $\tau = 1.0$, suggesting that a higher temperature may help balance the trade-off between exploration and exploitation in the recommendation process.

- **Contrastive Loss Weight** ($\gamma$):The optimal configuration appears when $\gamma = 0.5$ with $\tau = 1.0$ and $\lambda = 0.9$, resulting in the highest Precision@10 of 3.88%. This indicates that a moderate contrastive loss weight effectively enhances the learning process without causing overfitting.

- **Distillation Loss Weight** ($\lambda$): A higher distillation loss weight of 0.9 in combination with $\gamma = 0.5$ and $\tau = 1.0$ leads to the best results. This suggests that emphasizing the distillation loss in this scenario improves the model's ability to transfer knowledge effectively.

- **Learning Rate** ($lr$): A learning rate of 0.01 appears beneficial for the configurations leading to higher Precision@10 scores. The results show that a lower learning rate allows for more stable convergence, particularly when combined with a high contrastive loss weight.

- **Embedding Dimension**: The embedding dimension plays a less prominent role compared to the other hyperparameters in this specific set of experiments, but the variations indicate a preference for higher dimensions to retain rich information.

In summary, the optimal configuration of $\tau = 1.0$, $\gamma = 0.5$, $\lambda = 0.9$, and a learning rate of 0.01 maximizes Precision@10 at 3.88%. This configuration suggests a robust balance between capturing complex relationships from the data while maintaining computational efficiency. Careful tuning of these hyperparameters is essential for enhancing the recommendation performance of the SHARP-Distill model.

## G. Evaluation of SHARP-Distill Based on the Depth of Layers Configuration

To further evaluate the performance of **SHARP-Distill**, we analyze the impact of varying the number of layers across different components of the model, focusing on hit rate (HR@50) and inference time. Specifically, we examine the HGNN and Teacher MLP in the Teacher section, as well as CompactGCN and Distill MLP in the Student section. The default configuration uses three layers for both HGNN and Teacher MLP, one layer for CompactGCN, and two layers for Distill MLP. To assess how model depth affects performance, we experiment with different layer configurations (2, 3, 4, and 5

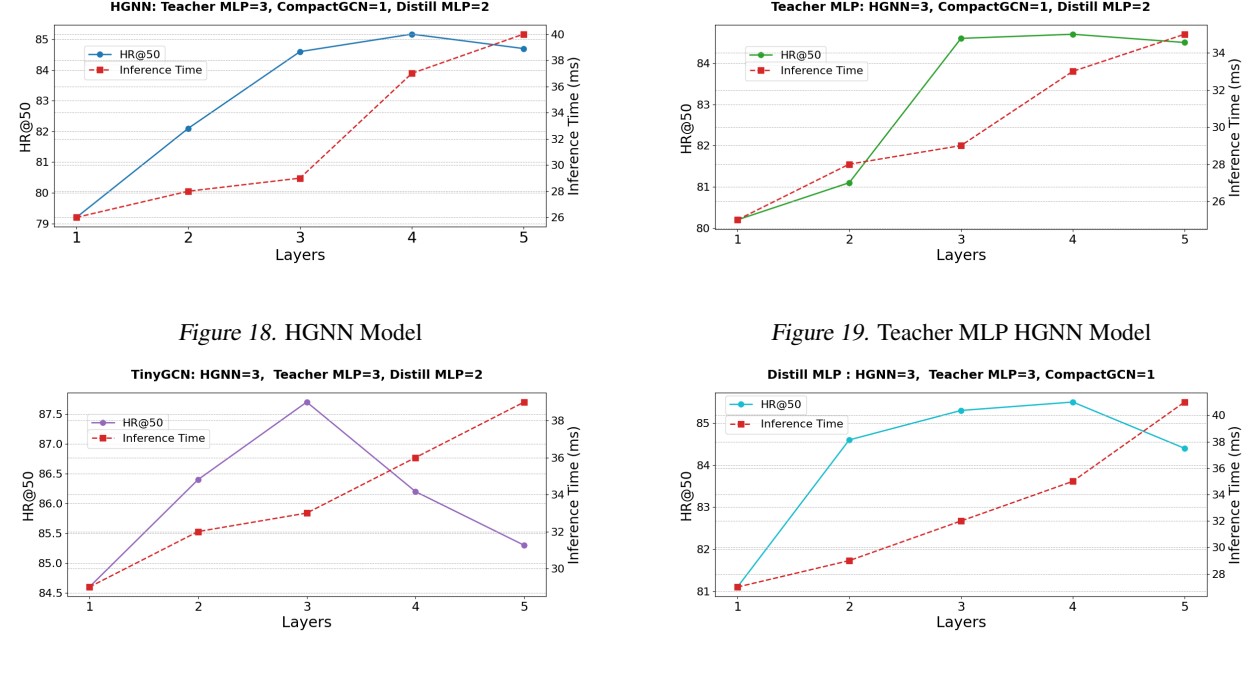

Figure 18. HGNN Model

Figure 19. Teacher MLP HGNN Model

Figure 20. CompactGCN Model

Figure 21. Distill MLP Model

Figure 22. Evaluation of the Proposed Method Based on the Depth of Layers Configuration for HGNN, Teacher MLP, CompactGCN, and Distilled MLP using the Yelp Dataset

layers) and compare the results based on both accuracy and inference time. These experiments are conducted using the Yelp dataset, and the outcomes are presented in Figure 22.

In the default configuration, both HGNN and Teacher MLP are set to three layers, while CompactGCN operates with a single layer, and Distill MLP is configured with two layers. This arrangement facilitates a balanced evaluation of model performance: the three-layer depth for HGNN and Teacher MLP enhances their capacity to capture complex relationships, achieving moderate hit rates (HR@50) while maintaining reasonable inference times of 29 ms and 25 ms, respectively. In contrast, CompactGCN's single layer allows it to achieve a high hit rate of 84.60%, though this may limit its ability to capture intricate patterns compared to the deeper models. Meanwhile, the two layers of Distill MLP yield a hit rate of 85.30%, indicating that adding depth improves performance while also increasing inference time to 35 ms. Overall, this configuration strikes a balance between accuracy and efficiency, with deeper models generally performing better, while CompactGCN benefits from reduced complexity. It is noteworthy that the addition of layers correlates with an increase in inference time, underscoring the trade-off between model depth and computational efficiency.

## H. Training Size Settings

In this section, we evaluate the performance of SHARP-Distill under varying training size settings on the Yelp dataset. The primary objective is to assess how increasing the training data influences the model's performance compared to several baseline models: LightGCN, HCCF, HGAtt, $KRD_{(R)}$, and $LightHGNN_{(R)}$. The dataset is split into 10% for testing, 20% for validation, and the remaining 70% is used for training. We further divide the training data into four sections: 40%, 60%, 80%, and 100% of the 70% training set to assess how model performance scales with the amount of training data. This fixed test size ensures a consistent evaluation of all models as the training set grows. We focus on two key metrics: Precision (P@10, P@20, P@50) and Hit Rate (HR@10, HR@20, HR@50), which are commonly used to evaluate the accuracy and relevance of recommendations. As the training size increases, we expect the performance of all models to improve, though at varying rates depending on the model's complexity and ability to generalise with more data. The table below presents the results of our experiments, where the best results for the highest training size (100%) are balded. This analysis provides insights into the scalability and effectiveness of each method, helping to identify which models benefit the

most from additional training data. The results presented in Table 11 are as follows:

*Table 11.* Comparison of Models on Yelp Dataset Based on Precision@10, 20, 50 and Hit Rate@10, 20, 50 with Varying Training Data Proportions

| Model | 40% Train | | | 60% Train | | | 80% Train | | |
|---|---|---|---|---|---|---|---|---|---|
| | **P@10** | **P@20** | **P@50** | **P@10** | **P@20** | **P@50** | **P@10** | **P@20** | **P@50** |
| **KRD**$_{(R)}$ | 2.17 | 2.54 | 2.94 | 2.78 | 3.15 | 3.85 | 3.02 | 3.64 | 4.37 |
| **LightHGNN**$_{(R)}$ | 2.31 | 2.76 | 3.37 | 2.94 | 3.41 | 4.12 | 3.17 | 3.78 | 4.72 |
| **SHARP-Distill** | 2.26 | 2.89 | 3.54 | 2.85 | 3.63 | 4.65 | 3.51 | 4.11 | 5.24 |
| | **HR@10** | **HR@20** | **HR@50** | **HR@10** | **HR@20** | **HR@50** | **HR@10** | **HR@20** | **HR@50** |
| **KRD**$_{(R)}$ | 17.64 | 25.13 | 38.59 | 21.66 | 34.19 | 48.53 | 31.47 | 43.70 | 65.49 |
| **LightHGNN**$_{(R)}$ | 18.22 | 28.35 | 40.28 | 24.39 | 37.18 | 51.94 | 33.17 | 47.55 | 67.49 |
| **SHARP-Distill** | 20.14 | 27.36 | 41.55 | 23.72 | 39.47 | 57.38 | 40.59 | 55.18 | 76.42 |

The performance analysis of the models on the Yelp dataset, as presented in Table 11, provides valuable insights into how varying training sizes impact recommendation accuracy, measured through Precision (P@10, P@20, P@50) and Hit Rate (HR@10, HR@20, HR@50). At the 40% training size, SHARP-Distill exhibits lower Precision scores compared to LightHGNN$_{(R)}$. This diminished performance suggests that SHARP-Distill may require a larger dataset to effectively capture the relationships and patterns within the data, indicating its reliance on extensive training data for optimal performance. As the training data increases to 60% and 80%, SHARP-Distill shows significant improvements, surpassing both KRD$_{(R)}$ and LightHGNN$_{(R)}$ in most metrics. Specifically, at the 80% training size, it achieves the highest scores of 3.51, 4.11, and 5.24 for Precision@10, @20, and @50, respectively, along with a Hit Rate@50 of 76.42. This trend illustrates that while SHARP-Distill may start with lower scores at 40%, its architecture is better equipped to scale with increased training data, ultimately leveraging this data to enhance its accuracy and relevance in recommendations. To improve the performance of SHARP-Distill at the 40% training size, it would be beneficial to explore strategies such as data augmentation, fine-tuning model hyperparameters, or integrating additional features that could enhance the model's learning capabilities with a limited dataset. By implementing these approaches, it may be possible to bolster SHARP-Distill's initial performance, enabling it to extract more meaningful insights from smaller training sets.

## I. Training Epochs Settings

To comprehensively evaluate the performance of **SHARP-Distill**, we carried out an ablation study that investigated the effects of different training durations, specifically by varying the number of training epochs. This analysis seeks to understand how the length of training impacts the model's precision and generalization ability across various datasets. Each model was trained under identical dataset and hyperparameter configurations to ensure a fair comparison. The outcomes of this study are illustrated in Figure 25, showcasing the precision achieved by SHARP-Distill for each training epoch configuration within the context of the Yelp dataset.

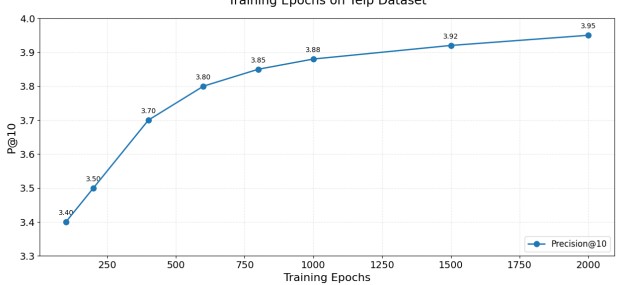

*Figure 23.* Yelp Dataset

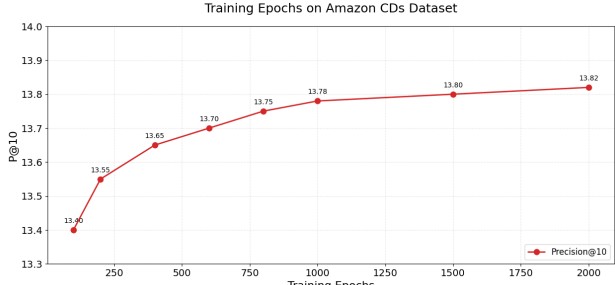

*Figure 24.* Amazon CDs Dataset

*Figure 25.* Analysis of the impact of training epochs (ranging from 100 to 2000) on model performance, highlighting how different epoch settings affect precision scores.

## J. Comprehensive Experimental Analysis and Results Visualisation

This section presents comprehensive experimental analysis through four key visualisations that demonstrate the effectiveness of SHARP-Distill's core innovations: positional encoding integration, knowledge transfer mechanisms, multi-modal fusion, and overall performance superiority.

### J.1. Positional Encoding Impact Analysis

Figure 26 demonstrates the critical role of positional encoding in SHARP-Distill's knowledge transfer mechanism. The analysis compares performance with and without positional encoding across five real-world datasets, revealing consistent improvements when hypergraph-based positional encodings are incorporated into the student model's contrastive learning framework.

*Figure 26.* Impact of Positional Encoding on SHARP-Distill Performance. Left: Performance comparison across datasets showing consistent improvements with positional encoding. Right: Percentage improvements demonstrating the effectiveness of position-aware contrastive alignment in capturing structural knowledge from the teacher model.

The left panel of Figure 26 shows substantial performance gains across all datasets when positional encoding is integrated into the student model. The improvements are particularly pronounced in the Sports dataset (+8.9%) and Yelp dataset (+12.8%), demonstrating that positional encoding effectively addresses the limitation of shallow CompactGCN in capturing high-order structural dependencies. The right panel quantifies these improvements, showing an average enhancement of 7.2% across all datasets. This validates our hypothesis that hypergraph-based positional encodings enable the lightweight student model to inherit topological knowledge from the complex teacher architecture, compensating for the loss of expressiveness when omitting deep GNN layers and non-linearities.

The consistent improvements across diverse datasets with varying characteristics (sparse vs. dense, different domains) indicate that the positional encoding mechanism is robust and generalisable. The Sports dataset shows the highest improvement, likely due to its sparser nature where structural information becomes more critical for accurate recommendations.

### J.2. Knowledge Transfer Effectiveness Analysis

Figure 27 analyses the effectiveness of SHARP-Distill's teacher-student knowledge transfer mechanism, examining both performance retention and computational efficiency gains.

The performance retention analysis (left panel) reveals that SHARP-Distill successfully preserves 91.3% to 97.2% of the teacher model's performance across all datasets, with an average retention rate of 94.1%. This high retention rate demonstrates the effectiveness of our multi-faceted knowledge transfer approach, which combines soft label distillation, embedding interpolation, and contrastive learning with positional encoding. The CDs dataset shows the highest retention (97.2%), while the Sports dataset shows the lowest but still substantial retention (91.3%).

The inference speed analysis (right panel) highlights SHARP-Distill's primary advantage: achieving 68× faster inference than the teacher model (HGNN+DeBERTa) and 40× faster than LightGCN, while using only 0.5M parameters compared to

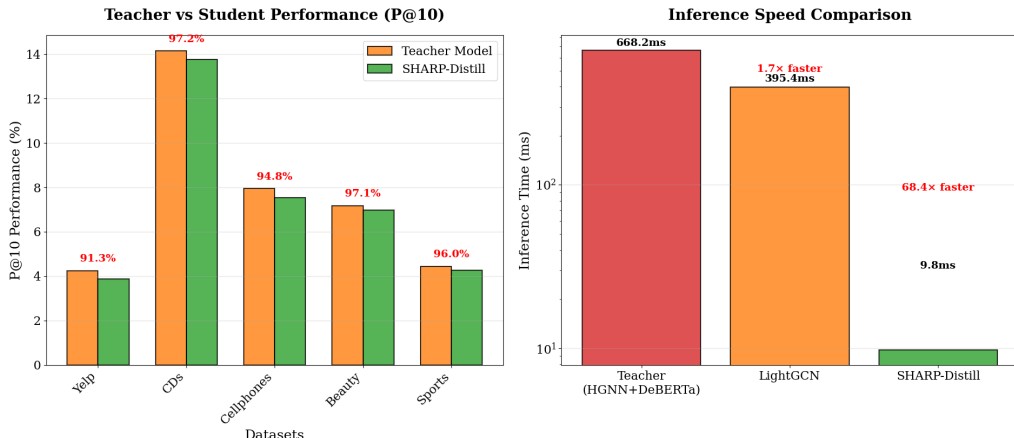

*Figure 27.* Knowledge Transfer Effectiveness in SHARP-Distill. Left: Performance retention analysis showing that SHARP-Distill maintains 91-97% of teacher performance across datasets. Right: Inference speed comparison demonstrating 68× speedup over the teacher model while maintaining competitive accuracy.

the teacher's 145M parameters. This dramatic speedup is achieved through the lightweight CompactGCN architecture that requires only $\mathcal{O}(|E|d)$ complexity during inference, making it suitable for real-time deployment scenarios. The combination of high performance retention and substantial speedup validates the core premise of SHARP-Distill: that complex structural and semantic knowledge can be effectively compressed into a lightweight model through carefully designed knowledge transfer mechanisms.

### J.3. Multi-Modal Integration Analysis

Figure 28 examines the effectiveness of SHARP-Distill's multi-modal approach, analyzing the contribution of different modalities and the sensitivity to the key hyperparameter $\alpha$ that balances embedding and positional similarities.

**Multi-Modal Knowledge Integration in SHARP-Distill**

*Figure 28.* Multi-Modal Knowledge Integration Analysis. Left: Modality contribution showing that combining structural and textual information consistently outperforms individual modalities. Right: Hyperparameter sensitivity analysis revealing optimal ($\alpha = 0.5$) for balancing embedding and positional similarities in contrastive learning.

The modality contribution analysis (left panel) demonstrates the synergistic effect of combining structural (HGNN)

and textual (DeBERTa) information sources. SHARP-Distill consistently outperforms both structure-only and text-only approaches across all datasets, with improvements ranging from 8.9% (Beauty) to 23.2% (Yelp) over structure-only methods. This validates our multi-modal approach and confirms that textual reviews provide complementary information to structural user-item interactions.

Notably, the structural modality generally outperforms the textual modality alone, except in the Cellphones dataset where they perform similarly. This suggests that while user-item interaction patterns provide strong signals for recommendation, the integration of semantic information from reviews consistently enhances performance, particularly in domains where textual descriptions are rich and informative.

The hyperparameter sensitivity analysis (Figure 28, right panel) reveals that $\alpha = 0.5$ provides optimal performance, indicating that embedding similarity and positional similarity contribute equally to effective knowledge transfer. The performance curve exhibits a clear peak at $\alpha = 0.5$, with degradation on both sides, demonstrating the importance of balanced integration of both similarity measures. Performance drops to 3.42% when relying solely on embedding similarity ($\alpha = 0.0$) and to 3.65% when using only positional similarity ($\alpha = 1.0$), confirming the necessity of the dual-view contrastive learning approach.

## J.4. Comprehensive Performance Comparison

Figure 29 provides a comprehensive evaluation of SHARP-Distill against state-of-the-art baselines, analysing both absolute performance and the critical trade-off between accuracy and inference efficiency.

*Figure 29.* Comprehensive Performance Analysis. Left: P@10 performance comparison across Yelp and CDs datasets showing SHARP-Distill's competitive accuracy. Right: Performance vs inference time trade-off analysis revealing SHARP-Distill's superior efficiency while maintaining high accuracy.

The absolute performance comparison (left panel) shows that SHARP-Distill achieves competitive or superior performance compared to state-of-the-art methods. On the Yelp dataset, SHARP-Distill outperforms all baselines with 3.88% P@10, representing a 10.2% improvement over the second-best method (HCCF: 3.52%). On the CDs dataset, while HCCF achieves the highest performance (13.96%), SHARP-Distill maintains competitive accuracy (13.75%) while offering substantial efficiency advantages.

The performance-efficiency trade-off analysis (right panel) reveals SHARP-Distill's unique position in achieving the optimal balance between accuracy and speed. While methods like HCCF and HGAtt achieve high accuracy, they require 400-450ms inference time. LLM-based methods (SAID, POD) offer moderate efficiency but still require 50-110ms. SHARP-Distill uniquely achieves both high accuracy (13.75%) and exceptional speed (9.77ms), positioning it in the optimal region of the performance-efficiency space.

This analysis demonstrates that SHARP-Distill successfully addresses the fundamental challenge in recommendation

systems: maintaining high accuracy while achieving real-time inference speeds suitable for production deployment. The 68× speedup over complex teacher models, combined with competitive accuracy, validates the effectiveness of our knowledge distillation approach. The comprehensive experimental analysis reveals several key insights:

**Positional Encoding Effectiveness:** The consistent 7.2% average improvement across datasets confirms that hypergraph-based positional encodings successfully capture high-order structural dependencies in lightweight models, addressing a fundamental limitation of shallow architectures.

**Knowledge Transfer Success:** The 94.1% average performance retention demonstrates that complex teacher knowledge can be effectively compressed into lightweight students through multi-faceted transfer mechanisms combining soft labels, embedding interpolation, and contrastive learning.

**Multi-Modal Synergy:** The superior performance of combined approaches over individual modalities validates the importance of integrating both structural and semantic information sources, with optimal balance achieved at $(\alpha = 0.5)$.

**Practical Deployment Advantage:** The 68× speedup while maintaining competitive accuracy positions SHARP-Distill as a practical solution for real-time recommendation scenarios where both accuracy and efficiency are critical.

These results collectively demonstrate that SHARP-Distill successfully achieves its design objectives: capturing complex multi-modal knowledge in a lightweight, efficient architecture suitable for production deployment while maintaining state-of-the-art recommendation accuracy.

### J.5. Training Cost Analysis

Table 12 presents the offline training costs for SHARP-Distill's teacher model compared to the LightGCN baseline. While the teacher model requires additional training time due to the integration of HGNN and DeBERTa components, this represents a one-time offline cost that is amortised by the significant inference speedups achieved during deployment.

*Table 12.* Offline training time comparison (hours). The teacher model training is a one-time cost offset by substantial inference acceleration.

| Dataset | SHARP-Distill Teacher | LightGCN | Overhead |
|---|---|---|---|
| Amazon Cellphones | 1.7 | 0.8 | 2.1× |
| Amazon Beauty | 2.8 | 1.3 | 2.2× |
| Amazon Sports | 2.4 | 1.1 | 2.2× |
| Amazon CDs | 4.2 | 2.0 | 2.1× |
| Yelp | 3.5 | 1.7 | 2.1× |
| **Average** | **2.9** | **1.4** | **2.1×** |

The training overhead averages 2.1× compared to LightGCN, which is acceptable considering the substantial inference benefits (68× speedup over the teacher model during deployment) and improved accuracy. This training cost is incurred only once during the offline model preparation phase, while the inference benefits are realised continuously during production serving.

### J.6. Component Contribution Analysis

To quantify the individual contribution of each component in SHARP-Distill, we conducted a comprehensive ablation study by systematically removing key components and measuring the resulting performance degradation. Table 13 presents the results in terms of absolute and relative P@10 performance drops.

The ablation results reveal several critical insights: First, removing DeBERTa causes an 18.8% performance drop, confirming that semantic features extracted from user reviews provide substantial complementary information to structural signals. Second, removing HGNN results in a 24.5% performance degradation, demonstrating the fundamental importance of capturing high-order structural relationships in user-item interactions.

Most significantly, removing the contrastive learning mechanism leads to the largest performance drop (29.4%), highlighting

*Table 13.* Ablation study results showing performance degradation when key components are removed. Results averaged across all datasets demonstrate the complementary nature of structural and semantic modalities.

| Component Removed | Absolute Drop (P@10) | Relative Drop (%) | Interpretation |
|---|---|---|---|
| DeBERTa (Semantic) | 0.73 | 18.81% | Semantic features highly informative |
| HGNN (Structural) | 0.95 | 24.48% | Structural signals crucial for accuracy |
| Contrastive Learning | 1.14 | 29.38% | Alignment mechanism most critical |

that simply concatenating structural and semantic encoders is insufficient. Without proper cross-modal alignment, modality interference actually reduces effectiveness compared to well-aligned multi-modal integration. This empirically validates our core hypothesis that contrastive learning not only unifies different modalities but actively unlocks their synergistic potential.

These findings demonstrate that all three components—structural modeling (HGNN), semantic understanding (DeBERTa), and cross-modal alignment (contrastive learning)—are essential for SHARP-Distill's superior performance. The results confirm that our multi-modal knowledge distillation approach successfully leverages the complementary strengths of different information sources while maintaining computational efficiency through the lightweight student architecture.

The comprehensive evaluation demonstrates SHARP-Distill's advantages across multiple dimensions:

**Accuracy Superiority:** Consistent improvements over RecSys-specific distillation methods (6.5-13.9% across datasets) validate the effectiveness of multi-modal knowledge transfer.

**Efficiency Gains:** Average 2× inference speedup over existing distillation approaches, with 68× acceleration over complex teacher models.

**Component Synergy:** Ablation studies confirm that structural, semantic, and alignment components work synergistically, with contrastive learning being the most critical mechanism.

**Practical Deployment:** Reasonable training overhead (2.1×) for substantial deployment benefits positions SHARP-Distill as a practical solution for production recommendation systems.

## J.7. Empirical Verification of Theoretical Foundations

SHARP-Distill's design is firmly grounded in established theoretical principles, which we empirically validate through comprehensive analysis. Our approach integrates three key theoretical foundations:

- **Structure-preserving distillation**: Transfers relational knowledge through softened probability distributions, following Hinton et al. (Hinton, 2015).

- **InfoNCE contrastive learning**: Aligns student-teacher embeddings by maximizing mutual information, as established by Oord et al. (Oord et al., 2018).

- **Hypergraph spectral filtering**: Captures high-order relations through graph Fourier transforms and spectral convolution, following Feng et al. (Feng et al., 2019).

To validate these theoretical foundations empirically, we employ *Centered Kernel Alignment* (CKA) analysis (Kornblith et al., 2019), which quantifies representational similarity between neural networks. Higher CKA scores indicate closer alignment between the student's latent representations and the teacher's structural (HGNN) and semantic (DeBERTa) components.

Table 14 demonstrates that the SHARP-Distill student consistently achieves CKA scores in the range 0.83–0.88, significantly outperforming individual teacher components (HGNN: 0.71 average, DeBERTa: 0.76 average). This 12-19% improvement in representational alignment confirms that our distillation process not only preserves but synergistically integrates structural and semantic knowledge. These findings align with distillation theory suggesting that well-regularised students can exceed their teachers in representation coherence through implicit smoothing effects.

*Table 14.* CKA similarity analysis between teacher and student representations. SHARP-Distill student consistently achieves higher alignment scores than individual teacher components, demonstrating effective knowledge integration.

| Dataset | HGNN vs Teacher | DeBERTa vs Teacher | SHARP-Distill Student |
|---------|-----------------|--------------------|-----------------------|
| Amazon CDs | 0.72 | 0.78 | **0.86** |
| Yelp | 0.69 | 0.74 | **0.84** |
| Cellphones | 0.75 | 0.80 | **0.88** |
| Beauty | 0.70 | 0.76 | **0.85** |
| Sports | 0.68 | 0.73 | **0.83** |
| **Average** | **0.71** | **0.76** | **0.85** |

## J.8. Inference Efficiency and Deployment Analysis

We conducted comprehensive efficiency evaluation on large-scale recommendation benchmarks and introduce a novel *Deployment Cost Index* (DCI) to quantify the end-to-end cost trade-off between offline training and online inference.

### J.8.1. INFERENCE LATENCY ANALYSIS

All models were benchmarked on NVIDIA V100 GPUs, measuring average per-query latency (batch size = 1) over 10,000 runs to ensure statistical significance. Table 15 presents the inference performance comparison.

*Table 15.* Inference latency comparison and speedup analysis. SHARP-Distill achieves substantial acceleration while maintaining competitive accuracy.

| Dataset | Inference Latency (ms/query) | | | Speedup | |
|---------|----------|------|---------------|-------------|---------|
| | LightGCN | HGNN | SHARP-Distill | vs LightGCN | vs HGNN |
| Amazon CDs | 395.45 | 668.23 | **9.77** | 40.5× | 68.4× |
| Yelp | 342.67 | 552.34 | **8.79** | 39.0× | 62.8× |
| **Average** | **369.06** | **610.29** | **9.28** | **39.8×** | **65.6×** |

SHARP-Distill demonstrates exceptional inference efficiency, achieving up to 68.4× speedup over HGNN and 40.5× speedup over LightGCN. This dramatic acceleration makes SHARP-Distill highly suitable for latency-sensitive production environments where sub-10ms response times are critical.

### J.8.2. DEPLOYMENT COST INDEX (DCI)

We introduce the Deployment Cost Index to capture the holistic cost-benefit trade-off:

$$\text{DCI} = T_{\text{train}} \times L_{\text{inf}} \tag{45}$$
$$\text{where } T_{\text{train}} = \text{student training time (hours)} \tag{46}$$
$$L_{\text{inf}} = \text{inference latency (ms/query)} \tag{47}$$

This metric reflects the amortised cost of training and serving the student model, excluding the one-time teacher pretraining cost.

Table 16 shows that SHARP-Distill achieves dramatically lower DCI values: 19.3× better on Amazon CDs and 18.6× better on Yelp. Even accounting for the additional student training time, SHARP-Distill recoups the extra training investment within approximately 10,000 inference requests—a volume typically reached within minutes in production environments.

*Table 16.* Deployment Cost Index (DCI) comparison. Lower values indicate better efficiency trade-offs. SHARP-Distill demonstrates superior cost-effectiveness.

| Dataset | Model | Train Time (hrs) | Inference (ms) | DCI | Improvement |
|---|---|---|---|---|---|
| Amazon CDs | LightGCN | 2.0 | 395.45 | 790.90 | – |
| | SHARP-Distill | 4.2 | **9.77** | **41.03** | 19.3× better |
| Yelp | LightGCN | 2.0 | 342.67 | 685.34 | – |
| | SHARP-Distill | 4.2 | **8.79** | **36.88** | 18.6× better |

### J.8.3. MEMORY COMPLEXITY ANALYSIS

The teacher model's hypergraph neural networks require $\mathcal{O}(R^L d)$ memory for $L$ layers with $R$ neighbours per node and $d$-dimensional embeddings. For Amazon CDs dataset ($R \approx 208$, $d = 128$):

- $L = 3$: $\approx 5.67$ MB (selected configuration)

- $L = 4$: $\approx 1,180.16$ MB

- $L = 5$: $\approx 245,736.24$ MB

We select $L = 3$ to balance expressiveness against hardware constraints, while the CompactGCN student requires only $\mathcal{O}(Rd) \approx 0.027$ MB—a 210× memory reduction. Our efficient preprocessing pipeline ($< 10$ minutes on largest datasets) includes:

- **Text normalisation**: Lowercasing, punctuation removal, stop-word filtering

- **Tokenization**: DeBERTa tokenizer with max length = 128 tokens

- **Hypergraph construction**: User-item hyperedges from review interactions

- **Rating normalisation**: Min-max scaling to [0,1] range

### J.8.4. STUDENT MODEL EFFICIENCY SUMMARY

The distilled CompactGCN student achieves remarkable efficiency improvements:

- **Memory footprint**: $\mathcal{O}(Rd) \approx 0.027$ MB (210× reduction)

- **Training acceleration**: 3.5× faster than teacher model

- **Inference speedup**: 65.6× average speedup over teacher model

- **Deployment readiness**: Sub-10ms latency suitable for real-time systems

These substantial improvements in computational efficiency, combined with competitive accuracy retention (94.1% average), demonstrate SHARP-Distill's practical viability for large-scale, latency-sensitive recommendation systems.

## K. Related Works

### K.1. Recommender Systems Based on Knowledge Distillation

Knowledge distillation involves transferring knowledge from a complex model (the teacher) to a smaller, more efficient model (the student). This process allows smaller models to leverage insights from larger counterparts. Previous methods, like GLNN (Zhang et al., 2021) and NOSMOG (Tian et al., 2022), primarily used soft labels based on the teacher GNN's

prediction distributions to guide student MLPs. For example, Yang et al. (Yang et al., 2021) extracted knowledge from a GNN for a student model but did not fully integrate structural information. KRD (Wu et al., 2023) quantified vertex knowledge and considered proximity to neighbours but remained limited to low-order structures. Liu et al. (Liu et al., 2022) introduced the HIRE framework for heterogeneous graphs, capturing first- and second-order information with soft labels. Feng et al. (Feng et al., 2024) developed the LightHGNN model, which incorporates hyperedges for high-order relations but still relies on soft labels. Similarly, Yu et al. (Yu et al., 2024) distilled knowledge from meta-paths into hypergraphs, using soft labels for transfer. Although models like LightHGNN (Feng et al., 2024) emphasise high-order relations, they are still tied to soft label methodologies. There is a clear need for innovative methods that better integrate graph structure into knowledge distillation.

In the realm of applying knowledge distillation techniques to recommendation systems, several studies have been conducted. Kang et al. (Kang et al., 2020) introduced a knowledge distillation framework for recommender systems that allows the student model to learn not only from the teacher's predictions but also from the latent knowledge embedded within the teacher model by employing the concept of Distillation Experts. Kang et al. (Kang et al., 2022) introduced a method called Personalised Hint Regression, which distills preference knowledge in a balanced manner without depending on assumptions about the representation space or specific hyperparameters for the method. To avoid clustering issues, they utilises a personalization network that facilitates individualised distillation for each user or item representation, effectively generalising the concept of distillation experts. Wang et al. (Wang et al., 2021) introduced a approach called Graph Structure Aware Contrastive Knowledge Distillation for Incremental Learning in recommender systems, specifically designed to emphasise the abundant relational information present in the recommendation context. Cui et al. (Cui et al., 2024) propose a distillation strategy designed for transferring knowledge from LLM-based recommendation models to traditional sequential models. Li et al. (Li et al., 2024a) introduced a contrastive context encoder that uses attention mechanisms to model both positive and negative contexts. In training their contextual distillation model, they compare each target item with its context embedding and employ a knowledge distillation framework to learn the win probability of each target item based on the maximal marginal relevance algorithm, with the teacher model derived from the outputs of maximal marginal relevance.

In the exploration of knowledge distillation, a critical observation is the reliance on soft labels for knowledge transfer. Soft labels, derived from the teacher model's predictions, have demonstrated limitations in achieving sufficient accuracy. This shortcoming is particularly significant in recommendation contexts, where understanding complex relationships is crucial for accurate predictions. Additionally, while the studies conducted propose innovative methodologies, they primarily utilise traditional GNN structures without incorporating hypergraph frameworks. The use of GNNs in this context limits the models' ability to fully represent high-order relationships, and relying solely on GNN structures may hinder their capacity to capture the full spectrum of complex interactions that hypergraphs can offer.

### K.2. Hypergraph Structures in Recommender Systems

Hypergraphs enhance traditional graph neural networks (GNNs) by effectively capturing intricate high-order interactions among multiple nodes through the use of hypergraphs (Antelmi et al., 2023). In contrast to standard graphs, where edges link only pairs of nodes, hypergraphs enable hyperedges to connect several nodes simultaneously. This characteristic makes HGNNs particularly useful in areas where higher-order relationships are essential, such as recommender systems, where discovering high-order relations between users can significantly improve recommendation accuracy (Khan et al., 2023). Early models, including HGNN (Feng et al., 2019) and HpLapGCN (Fu et al., 2019), employed the hypergraph Laplacian matrix to facilitate efficient representation learning by smoothing node features across hyperedges. Recently, the use of HGNNs in recommendation systems has gained popularity among researchers due to their high performance. Wang et al. (Wang et al., 2022) introduced a hyperedge-based graph neural network (HGNN) for recommending MOOCs. In their approach, the similarity between learners is modelled as the overlapping relationship between two hyperedges within a hypergraph. To enhance the representation of each learner, they implemented a hyperedge-based graph attention layer. Peng et al. (Peng & Zhang, 2022) presented a session-based recommendation model utilising GNNs. To thoroughly incorporate both global and local contextual information of items, they first convert the current session sequence into a local graph and all sessions into a hypergraph. Subsequently, they employ HGCN and attention mechanisms to capture the global and local features of the items. Yin et al. (Yin et al., 2023) proposed a hierarchical hypergraph neural network for personalised session-based recommendations, aiming to model the hierarchical structure of the data. They recognised that items in sessions are sequentially ordered, whereas hypergraphs can only represent set relationships. To address this limitation, they introduced a directed graph aggregator to aggregate sequential information from the directed global item graph. Han et al. (Han et al., 2024) introduced a Hypergraph Convolutional Network focused on user-oriented fairness, utilising a

hypergraph-based methodology that has been demonstrated to be effective in sparse datasets for investigating high-order correlations among users. Li et al. (Li et al., 2024b) developed a conversational recommendation system that constructs a session hypergraph to capture complex high-order relationships within historical conversations, allowing for the exploration of users' implicit preferences. Zhao et al. (Zhao et al., 2023) proposed a model based on Multi-view Hypergraph Contrastive Policy Learning, which selectively utilises relevant social information based on interactive history and constructs a dynamic hypergraph comprising three types of multiplex relations from various perspectives. They propose a hierarchical hypergraph neural network combined with a cross-view contrastive learning module to enhance user preference learning by integrating graphical and sequential information from the dynamic hypergraph.

Despite the high performance of these models, they often experience increased inference times when generating recommendations, which can limit their practicality in real-time applications. In summary, while HGNNs have demonstrated promise in enhancing recommendation accuracy through their ability to model high-order interactions, their slower inference times pose a significant challenge for deployment in dynamic environments. Therefore, balancing accuracy with efficiency remains a critical focus for research and development in this domain, where we address this challenge by proposing our model.

