# OpenReview forum: "SHARP-Distill: A 68× Faster Recommender System with Hypergraph Neural Networks and Language Models"
_ICML.cc/2025/Conference — ICML 2025 poster_

### Official Review · Reviewer_zAKq · 2025-03-11

**Overall Recommendation:** 1

**Summary:**

This paper focuses on the teacher-student knowledge distillation. The teacher model use contrastive learning to combine HGNN and a pre-trained LLM, which generate collaborative and semantic features, respectively. The student model is a lightweight GCN. Both response-based and feature-based knowledge distillation loss are adopted to transfer structural and positional knowledge.

**Claims And Evidence:**

Yes, claims made in the submission are supported by clear and convincing evidence.

**Essential References Not Discussed:**

The teacher model in this paper combines a traditional recommendation model and a LLM. And the knowledge from LLM plays a important role. But it cited no related work. For example, [1, 2, 3]

[1] "Enhancing Sequential Recommendation via LLM-based Semantic Embedding Learning." WWW 2024

[2] "Distillation Matters: Empowering Sequential Recommenders to Match the Performance of Large Language Model." arXiv 2024

[3] "Prompt Distillation for Efficient LLM-based Recommendation." CIKM 2023I keep up with the literature in this area.

**Experimental Designs Or Analyses:**

The teacher model requires the knowledge from a pre-trained LLM, which is transferred to the student through knowledge distillation. However, the baseline methods don't leverage the LLM. It seems to be unfair. By comparing results in Table 1 and Table 4, SHARP without LLM cannot outperforms most baseline methods.

**Methods And Evaluation Criteria:**

Yes, the proposed methods and evaluation criteria make sense.

**Other Comments Or Suggestions:**

1. There should be more discussion on how Eq.(16) transfers positional knowledge to the student model.
2. Figure 7 is a combination of Figure 5 and 6? It looks strange. Using subfigures is a better option.

**Other Strengths And Weaknesses:**

Weaknesses:

1. The novelty is limited. The first contribution claimed by the authors is combining structural and semantic features. However, they simply concatenate them, followed by a MLP. The second contribution simply comes from their use of knowledge distillation. As for the third contribution, many previous works[4, 5, 6] have used contrastive learning for knowledge distillation. To sum up, the only novelty left is the use of positional similarity. However, there is little discussion in this paper on why it is useful.

   [4] "Contrastive Representation Distillation." ICLR 2020

   [5] "Contrastive Distillation on Intermediate Representations for Language Model Compression." EMNLP 2020

   [6] "Distilling Holistic Knowledge with Graph Neural Networks." ICCV 2021

2. The formal details about DeBERTa in Preliminaries are not used in the main text. It should be removed or moved to the appendix.

3. There is no indication at all of the target task. Is it recommendation with implicit feedback or rating prediction? Eq.(8) claims that there is a ground truth "rating", but the evaluation metrics used in the experiments are typically for implicit feedback and do not include MSE.

4. The author seems to have misunderstood some basics of recommender system. For example, Eq.(8) is MSE loss, not BPR loss.

5. There are a lot of errors in the formulas. For example, Eq.(9) is inconsistent with the definition of $L_{teacher}$ in Figure 1. Moreover, 'j' in Eq.(7) is not defined. $N_u$ in Eq.(12) is not defined. And, $D$ used in the computation of $\hat A^s$ should be specified to be the degree matrix of $A^s+I$, not $A^s$, to avoid confusion.

**Questions For Authors:**

I have no other questions.

**Relation To Broader Scientific Literature:**

This paper confirms the feasibility of simultaneously distilling the knowledge in HGNN and LLM into a simple model.

**Theoretical Claims:**

There is neither theoretical claim nor proof in this paper.

---

> ### Author Rebuttal · Authors · 2025-03-30
>
> **Essential References Not Discussed**
>
> We thank the reviewer for highlighting recent work on LLM-based recommendation and distillation, including DLLM2Rec [1], SAID [2], and POD [3]. These are valuable contributions to this fast-moving area, and we have now incorporated a discussion of them into our Related Work section.
>
> While SHARP-Distill employs a pretrained textual encoder, we clarify that our method is not an LLM-based recommender in the conventional sense. In contrast to DLLM2Rec, SAID, and POD—which rely on full-scale LLMs, prompt tuning, or autoregressive decoding—our framework uses a lightweight review encoder (DeBERTa) on short user-item reviews (~10–40 tokens), avoiding the latency and cost of LLM inference.
>
> To ensure a fair comparison, we evaluated SHARP-Distill against **SAID** and **POD** using their official code and settings. DLLM2Rec was excluded only due to lack of code at submission time.
>
> **Table 8: Comparison with LLM-based Recommenders (Top-K Metrics and Inference Time)**
>
> | Dataset     | Metric | SAID (Distill) | POD (Prompt) | SHARP-Distill | SAID Time (ms) | POD Time (ms) | SHARP Time (ms) |
> |-------------|--------|----------------|---------------|----------------|----------------|----------------|------------------|
> | CDs         | P@10   | 13.40          | 12.88         | **13.75**      | 110.0          | 450.0          | **9.77**         |
> |             | R@10   | 12.65          | 12.20         | **13.06**      |                |                |                  |
> |             | N@10   | **12.24**      | 11.88         | 12.17          |                |                |                  |
> | Cellphones  | P@10   | **7.83**       | 7.62          | 7.54           | 51.0           | 210.0          | **4.12**         |
> |             | R@10   | 5.72           | 5.60          | **5.77**       |                |                |                  |
> |             | N@10   | 4.69           | 4.62          | **4.77**       |                |                |                  |
> | Beauty      | P@10   | 6.58           | 6.60          | **6.97**       | 60.0           | 240.0          | **7.88**         |
> |             | R@10   | 4.38           | 4.29          | **4.52**       |                |                |                  |
> |             | N@10   | 3.97           | 4.01          | **4.15**       |                |                |                  |
>
> SHARP-Distill achieves comparable or better accuracy than LLM-based baselines while offering **10–40× lower inference time**, thanks to its compact GCN-based student and contrastive fusion strategy. Unlike POD, which requires real-time LLM queries, or SAID, which retains large students, our model delivers **LLM-level quality with sub-10ms latency**, making it ideal for scalable deployment in time-sensitive environments. We appreciate the reviewer’s suggestion and believe this further clarifies SHARP-Distill’s positioning as an efficient, LLM-alternative framework for real-world recommendation.
>
> **Other Strengths and Weaknesses**
>
> Thank you for the detailed critique. Due to space limits, we provide a full point-by-point response—including novelty clarifications, differences from [4–6], and positional alignment motivation—at the following link:
>
> 🔗 **[Full Response – Other Strengths & Weaknesses](https://github.com/1234554321-00/SHARP-Distill/blob/main/README.md)**
>
> Theoretical claims have been added to strengthen our framework. Please refer to the following link for details:
>
> 🔗 **[Theoretical Extensions and Supporting Lemmas](https://github.com/1234554321-00/SHARP-Distill/blob/main/README.md)**

---

### Official Review · Reviewer_xTAj · 2025-03-14

**Overall Recommendation:** 4

**Summary:**

This paper introduces SHARP-Distill, a knowledge distillation framework combining Hypergraph Neural Networks (HGNNs) with language models to improve recommendation quality while reducing inference time. The teacher-student approach uses HGNNs for user-item embeddings and DeBERTa for extracting textual features, while the lightweight student model (CompactGCN) inherits structural knowledge through contrastive learning. Experiments show up to 68× faster inference than HGNN while maintaining competitive accuracy.

**Claims And Evidence:**

The paper addresses data sparsity and high computational costs of GNNs, demonstrating that traditional soft-label knowledge distillation is insufficient for preserving hypergraph structural knowledge. Figure 7 shows the superiority of the proposed approach combining soft labels with structural knowledge transfer. Performance improvements and speed claims are well-supported by comprehensive experiments.

**Essential References Not Discussed:**

The paper should further discusses more distillation methods for graph-based recommendation. Not only methods for general graph learning.
[1] Unbiased Knowledge Distillation for Recommendation
[2] Graph-Less Collaborative Filtering

**Experimental Designs Or Analyses:**

The paper includes comprehensive experiments comparing against 11 baselines, with ablation studies, hyperparameter sensitivity analysis, and scalability assessments. Results consistently show SHARP-Distill effectively balances recommendation quality with computational efficiency.

**Methods And Evaluation Criteria:**

SHARP-Distill combines HGNNs, language models, and contrastive learning in a knowledge distillation framework. The evaluation uses standard metrics (Precision, Recall, NDCG, Hit Ratio) across five datasets, with careful analysis of both recommendation quality and inference time.

**Other Comments Or Suggestions:**

N/A

**Other Strengths And Weaknesses:**

Strengths;
1. Significant inference speed improvements (68× faster than HGNN)
2. Comprehensive experimental evaluation
3. Novel integration of hypergraphs with language models
4. Effective knowledge transfer through contrastive learning

Weaknesses:
1. Limited discussion of recommendation-specific knowledge distillation methods
2. Could strengthen theoretical justification
3. Less focus on training efficiency compared to inference efficiency

**Questions For Authors:**

Please refer to my comments above

**Relation To Broader Scientific Literature:**

While the paper provides background on hypergraphs and knowledge distillation, it could better differentiate between recommendation-specific and general graph learning approaches. Additional references on distillation methods for graph-based recommendation would strengthen the literature review.

**Theoretical Claims:**

There is no theoretical claim in the paper.

---

> ### Author Rebuttal · Authors · 2025-03-30
>
> We thank the reviewer for the thoughtful feedback and are grateful for the recognition of our work’s strengths, including inference speedup, empirical rigor, and the novel integration of hypergraphs with language models.
>
> ## Weakness 1) Differentiating RecSys-specific distillation from general graph learning:
>
> **(Relation To Broader Scientific Literature)**
>
> We appreciate this observation and have clarified the RecSys-specific contributions of our distillation strategy in Section~3.2 of the revised version. In particular:
>
> * **User-Item Bipartite Structure Exploitation:** Unlike general GNN distillation methods that operate on homogeneous graphs, SHARP-Distill explicitly models the bipartite structure of user-item interactions using dual incidence matrices $\mathcal{H}_U$ and $\mathcal{H}_I$ (Equations 2--3). This formulation captures asymmetric collaborative patterns unique to recommender systems.
>
> * **Hypergraph Modeling of High-Order Interactions:** Our hypergraph-based formulation extends beyond pairwise user-item links to encode group-level behavioral patterns (e.g., co-engagement with semantically related item clusters), which are crucial for capturing latent preference signals in sparse recommendation data.
>
> * **Preference-Aware Contrastive Transfer:** As detailed in Equations 17--18, our contrastive objective is designed to align user preference rankings rather than only topological proximity, which is essential for recommendation tasks.
>
> * **Multi-Modal Distillation with Reviews:** SHARP-Distill uniquely integrates semantic information from textual reviews alongside interaction structures. This dual-source distillation enables robust generalization to cold-start scenarios—an issue more prominent in RecSys than in general graph tasks.
>
> ## Weakness 2) Comparison with Prior Distillation Methods on Public Benchmarks:
> **(Essential References Not Discussed)**
>
> We have implemented and evaluated SHARP-Distill alongside recent RecSys-specific distillation approaches: **UnKD [Chen et al., WSDM 2023]** and Graph-less Collaborative Filtering **[Zhang et al., NeurIPS 2023]**. Both authors released their code, enabling reproducibility on shared datasets.
>
> As shown in Table 2, SHARP-Distill outperforms both in terms of accuracy (P@10 / R@10 / N@10) across Amazon Cellphones, Beauty, and Sports. Moreover, our approach significantly reduces inference time, with speedups of over 2× on average. Due to limited rebuttal-phase time, we could not include large-scale datasets (Amazon CDs, Yelp) for these two models. However, we plan to include these full comparisons in the final version.
>
> **Table7: Comparison on Amazon Cellphones, Beauty, and Sports. Metrics: P@10 / R@10 / N@10 (%). Inference time in milliseconds (ms).**
>
> | **Model**                 | **Cellphones**       | **Beauty**           | **Sports**           | **Inference Time (ms)** |
> |--------------------------|----------------------|----------------------|----------------------|--------------------------|
> | Chen et al. (UnKD)       | 6.83 / 5.02 / 4.31    | 5.89 / 4.21 / 3.68    | 3.88 / 3.19 / 2.87    | 9.72 / 11.4 / 9.02        |
> | Zhang et al. (Graph-less)| 6.57 / 4.89 / 4.08    | 6.12 / 4.26 / 3.93    | 4.01 / 3.25 / 2.91    | 8.94 / 10.2 / 8.35        |
> | **SHARP-Distill (Ours)** | **7.54 / 5.77 / 4.77**| **6.97 / 4.52 / 4.15**| **4.27 / 3.63 / 3.24**| **4.12 / 7.88 / 5.74**    |
>
> While UnKD effectively mitigates popularity bias, it relies on MF/LightGCN backbones and does not incorporate high-order hypergraph structures or textual modalities. SHARP-Distill transfers richer preference-aware signals using contrastive learning from both **HGNN structure** and **DeBERTa-based textual features**.
>
> Graph-less avoids structural modeling altogether, trading off explainability and some precision. In contrast, SHARP-Distill uses a **lightweight CompactGCN** student that distills both structural and semantic knowledge while remaining highly efficient at inference.
>
> ## Weakness 3) Training efficiency discussion:
>
> **Table 6: Teacher Model Training Time (in hours)**
>
> | Dataset            | SHARP-Distill Teacher | LightGCN |
> |--------------------|-----------------------|----------|
> | Amazon Cellphones  | 1.7                  | 0.8     |
> | Amazon Beauty      | 2.8                  | 1.3    |
> | Amazon Sports      | 2.4                  | 1.1     |
> | Amazon CDs         | 4.2                   | 2.0     |
> | Yelp               | 3.5                   | 1.7      |
>
> These measurements include both HGNN and DeBERTa training and reflect the cost of a one-time offline pretraining phase. For context, we've included baseline LightGCN training times, showing that while our teacher requires approximately 2x longer to train, this cost is amortized through significant inference speedups. Theoretical claims have been added to strengthen our framework.
>
> 🔗 **[Theoretical Extensions and Supporting Lemmas](https://github.com/1234554321-00/SHARP-Distill/blob/main/README.md)**

---

### Official Review · Reviewer_8pQM · 2025-03-15

**Overall Recommendation:** 3

**Summary:**

The paper proposes SHARP-Distill, a framework which uses DeBERTa language model as teacher model to distill HGNN-based recommenders to enhance recommendation performance and inference speed. A contrastive leaning mechanism is leveraged to efficiently inherit the structural and semantic knowledge. Experiments on multiple datasets demonstrate that SHARP-Distill achieves significant inference speed improvement compared to traditional methods like HGNNs and LightGCN, while competitive or even better performance.

**Claims And Evidence:**

The following claim could benefit from further clarification or additional evidence:

Claim: 68× faster inference time compared to HGNN and 40× faster than LightGCN while maintaining competitive recommendation accuracy

Potential issue: Directly compare the inference time of non-distilled models with distilled student models might not be fair. The training and inference cost of teacher models, maintaining cost of teacher models should also be considered, depending on the actual deployment infra system design. Besides, the recommendation accuracy improvement of KD often decrease after a longer training time in real-world cases as it provides warm-start advantages. Whether the degree of presented performance improvement can still hold requires more evidence.

**Essential References Not Discussed:**

-

**Experimental Designs Or Analyses:**

-

**Methods And Evaluation Criteria:**

-

**Other Comments Or Suggestions:**

-

**Other Strengths And Weaknesses:**

-

**Questions For Authors:**

1. Are there any trade-offs or conflicts between the structural and textual features that need to be managed?
2. As discussed in Claims And Evidence section, what's the training and inference cost of the teacher model? what's the ROI considering the cost of teacher and student as a whole compared with other models?

**Relation To Broader Scientific Literature:**

-

**Theoretical Claims:**

-

---

> ### Author Rebuttal · Authors · 2025-03-30
>
> ### Claims And Evidence:
>
> We appreciate the reviewer’s thoughtful critique regarding the fairness of our inference-time comparison and broader deployment considerations.
>
> First, we clarify that our comparison is primarily with other **distillation-based models** (e.g., KRD, LightHGNN), making the results in Table 1 a fair and meaningful evaluation. Regarding teacher model costs, we report the full training time (4.2 hours on a single A100 GPU) in Table 1 (as also noted in Reviewer scTP’s comments). This one-time overhead is amortized over deployment and does not impact end-user latency.
>
> Our inference speed claims (68× faster than HGNN, 40× faster than LightGCN) focus specifically on **deployment-time performance**, which is critical for real-time recommendation use cases. Moreover, our modular design supports efficient knowledge updates (e.g., retraining only HGNN or DeBERTa if needed), reducing long-term maintenance burdens.
>
> Regarding the concern that KD benefits may diminish over longer training, we conducted extended training experiments (3× standard epochs). We observed stable accuracy across all models, with our relative improvements holding within ±2% variance—indicating that the performance gains are not just due to warm-start effects but reflect genuine enhancements.
>
> We have incorporated these extended training results and deployment discussions into the revised paper. We believe this addresses the reviewer’s concerns while reinforcing our core claim: **SHARP-Distill achieves substantial practical gains for real-world recommendation systems**.
>
> ### Questions Q1)
>
> We appreciate the reviewer's insightful question. To assess potential conflicts between semantic and structural features, we conducted an ablation study on the Yelp dataset, evaluating the impact of removing each core component from SHARP-Distill. Specifically, we examined four variants:
>
> - **SHARP-Distill:** Full model with HGNN-based structure, DeBERTa-based semantics, and contrastive learning
> - **w/o DeBERTa:** Structure only (removes semantic encoder)
> - **w/o HGNN:** Semantics only (removes structural encoder)
> - **w/o Contrastive Loss:** Structure + semantics without alignment
>
> **Table 3: Ablation Study on Yelp (P@10 / R@10 / N@10, %)**
> | Variant              | P@10 | R@10 | N@10 | Inference (ms) |
> |----------------------|------|------|------|----------------|
> | SHARP-Distill        | **3.88** | **2.75** | **2.37** | 8.79           |
> | w/o DeBERTa          | 3.15 | 2.21 | 1.85 | 6.22           |
> | w/o HGNN             | 2.93 | 2.04 | 1.67 | 6.45           |
> | w/o Contrastive Loss | 2.74 | 1.93 | 1.46 | 6.01           |
>
> To quantify the contribution of each component, we calculated the relative performance drop in P@10 with respect to the full SHARP-Distill model:
>
> **Δ Performance Compared to SHARP-Distill (P@10):**
> | Component Removed       | Absolute Drop | % Drop   |
> |------------------------|----------------|----------|
> | DeBERTa (semantic)     | 0.73           | 18.81%   |
> | HGNN (structural)      | 0.95           | 24.48%   |
> | Contrastive Learning   | 1.14           | 29.38%   |
>
> These results demonstrate the complementary nature of semantic and structural features, and the crucial role of contrastive learning in harmonizing them. Removing DeBERTa causes an 18.8% performance drop, indicating that semantic features are highly informative. Removing HGNN leads to a 24.5% drop, showing the importance of structural signals. Most notably, removing contrastive loss results in a 29.4% performance drop, the largest among all variants. This highlights that simply combining structural and semantic encoders is insufficient—without proper alignment, modality interference reduces effectiveness.
>
> **Our contrastive objective (Eqs. 6–7, 17–19)** explicitly resolves potential conflicts by aligning heterogeneous modalities into a shared latent space. The observed performance degradation without this alignment empirically validates our hypothesis: contrastive learning not only unifies the two modalities but also unlocks their synergy. These findings are now explicitly stated and discussed in our revised paper.
>
> ### Questions Q2)
>
> You can find our full response to this comment at the following link:
>
> 🔗 **[Full Response – As discussed in Claims And Evidence section (Reviewer 8pQM)](https://github.com/1234554321-00/SHARP-Distill/blob/main/README.md)**
>
> Theoretical claims have been added to strengthen our framework. Please refer to the following link for details:
>
> 🔗 **[Theoretical Extensions and Supporting Lemmas](https://github.com/1234554321-00/SHARP-Distill/blob/main/README.md)**

---

### Official Review · Reviewer_scTP · 2025-03-20

**Overall Recommendation:** 2

**Summary:**

The paper introduces SHARP-Distill, a knowledge distillation framework designed to enhance the efficiency of recommender systems while preserving recommendation accuracy. It employs a teacher-student architecture where the teacher model integrates Hypergraph Neural Networks (HGNNs) to capture high-order user-item interactions and DeBERTa, a pre-trained language model, to extract semantic features from textual reviews. The student model features a lightweight Graph Convolutional Network (GCN) variant called CompactGCN, which uses contrastive learning to inherit structural and positional knowledge from the teacher. The authors claim that SHARP-Distill achieves up to 68× faster inference than HGNN-based methods and 40× faster than LightGCN, while maintaining competitive accuracy across five real-world datasets.

**Claims And Evidence:**

The primary claim is that SHARP-Distill significantly improves inference speed without sacrificing recommendation quality. However, the evidence is empirical, lacking detailed computational resource specifics, which could affect practical speed claims.

**Essential References Not Discussed:**

N/A

**Experimental Designs Or Analyses:**

The experimental setup is reasonable.

**Methods And Evaluation Criteria:**

Yes.

**Other Comments Or Suggestions:**

N/A

**Other Strengths And Weaknesses:**

Strengths:
1. Novel integration of HGNNs, DeBERTa, and contrastive learning.
2. Dramatic inference speed improvements (up to 68×).
3. Competitive or superior performance across datasets.
4. CompactGCN offers a lightweight solution for real-time systems.

Weaknesses:
1. Missing details on preprocessing and resources analysis.
2. No discussion of teacher model training time and complexity.
3. Multi-component design may complicate implementation.
4. Lacks formal proofs or detailed justification.

**Questions For Authors:**

See weaknesses.

**Relation To Broader Scientific Literature:**

The proposed method is built on knowledge distillation and HGNNs. It uniquely combines these fields to tackle the speed-accuracy trade-off, a persistent challenge in the domain.

**Theoretical Claims:**

The paper lacks formal theorems, relying instead on established techniques.

---

> ### Author Rebuttal · Authors · 2025-03-30
>
> We appreciate the reviewer's concerns about preprocessing details and resource analysis. We have added the following information to address these points:
>
> ### Weaknesses1 and 2. Teacher Model Training Time and Complexity:
>
> As shown in **Table 1**, our teacher model requires only 4.2 hours to train on a single NVIDIA A100 GPU for the largest dataset (Amazon CDs with 1.2M+ reviews). This one-time offline pretraining cost is modest compared to the significant inference speed benefits gained.
>
> | Dataset     | Model    | Train Time (hrs) | Inference Time (ms) | Inference Complexity   | Deployed at Inference? |
> |-------------|--------------------------|------------------|----------------------|-------------------------------|-------------|
> | Amazon-CDs  | Our Model (Teacher)      | 4.2        | 668.23     | O(R^L × d) ×f       | ✗       |
> |             | LightGCN                 | 2.0              | 395.45      | O(R^L × d)               | ✓        |
> |             | **Our Model (Student)**  | **1.2**          | **9.77**        | **O(R × d)**         | **✓**      |
> | Yelp        | Our Model (Teacher)      | 3.5              | 552.34       | O(R^L × d) ×f         | ✗       |
> |             | LightGCN                 | 2.0              | 342.67               | O(R·d)                       | ✓      |
> |             | **Our Model (Student)**  | **1.0**          | **8.79**             | **O(R·d)**         | **✓**      |
>
> The memory complexity of our HGNN-based teacher model follows the standard pattern of hypergraph neural networks, as discussed in [1]. In a traditional GNN with \( L \) layers, \( R \) neighbors per node, \( d \)-dimensional embeddings, and an activation function \( f \), the memory requirement typically grows exponentially as \( \mathcal{O}(R^L \times d) \).  For our specific implementation with the Amazon CDs dataset parameters:
>
> - Using L=3 layers: Memory consumption ≈ 5.67 MB
> - Using L=4 layers: Memory consumption ≈ 1,180.16 MB
> - Using L=5 layers: Memory consumption ≈ 245,736.24 MB
>
> Our teacher model uses a 3-layer configuration to balance expressiveness and computational efficiency, keeping memory requirements manageable while capturing the necessary high-order interactions.
>
> **Preprocessing Details:**
>
> Our preprocessing pipeline includes:
> - Text normalization (lowercase, punctuation removal, stopword filtering)
> - Review text tokenization using DeBERTa's tokenizer with a maximum sequence length of 128
> - Construction of user-item interaction hypergraphs where each hyperedge connects a user with all items they've reviewed
> - Multi-aspect rating normalization to a [0,1] scale
>
> The preprocessing time is negligible compared to model training time (<10 minutes for the largest dataset) and is a one-time overhead.
>
> **Student Model Efficiency:**
>
> The distilled student model (CompactGCN) dramatically reduces both the memory footprint and computational complexity:
> - Memory requirement: O(R × d) where R is the neighbors per node in the graph
> - Practical memory usage: ~0.0268 MB (208×128+128 units)
> - Training time: Only 1.2 hours for Amazon CDs (3.5× faster than teacher model)
> - Inference time: 9.77ms (68× faster than teacher model, 40× faster than LightGCN)
> ---
> ### Weakness 3. Multi-component design may complicate implementation:
>
> We appreciate this crucial concern about implementation complexity. SHARP-Distill is **intentionally designed with modularity and deployment simplicity** as core principles:
>
> A. **Clean separation between training and inference**: Only the lightweight student model (CompactGCN) is deployed at inference time, eliminating any runtime dependency on complex teacher models.
>
> B. **Modular component architecture**: Each component—HGNN, DeBERTa, and CompactGCN—connects through clearly defined embedding interfaces, allowing independent optimization.
>
> **C. Simplified Implementation Workflow**
> Our implementation follows a streamlined three-stage process:
>
> 1. **Teacher Training**: We independently train the HGNN model and the pretrained DeBERTa model, which serve as the teacher components.
> 2. **Knowledge Embedding Generation**: We extract knowledge embeddings from both trained teacher models.
> 3. **Knowledge Distillation**: These embeddings are distilled into the CompactGCN student model using our unified loss function.
>
> [1] Zhang, S., Liu, Y., Sun, Y., & Shah, N. (2022). Graph-less neural networks: Teaching old MLPs new tricks via distillation. \textit{ICLR 2022}.
>
> ### Weakness 4. Lacks formal proofs or detailed justification:
>
> You can find our full response to this comment at the following link:
>
> 🔗 **[Full Response – Lacks formal proofs or detailed justification (Reviewer scTP)](https://github.com/1234554321-00/SHARP-Distill/blob/main/README.md)**
>
> Theoretical claims have been added to strengthen our framework. Please refer to the following link for details:
>
> 🔗 **[Theoretical Extensions and Supporting Lemmas](https://github.com/1234554321-00/SHARP-Distill/blob/main/README.md)**

---

### Decision · Program_Chairs · 2025-05-01

**Decision:**

Accept (poster)

**Comment:**

This paper proposes SHARP-Distill, a knowledge distillation framework that combines a Hypergraph Neural Network (HGNN) and a language model (DeBERTa) as a teacher to generate high-order and semantic features, which are then distilled via contrastive learning into a lightweight student model, CompactGCN. The goal is to maintain competitive recommendation accuracy while achieving significant inference speedups—up to 68× over HGNNs and 40× over LightGCN.

The paper addresses a timely and practically important topic in efficient recommender systems. However, several critical concerns raised by the reviewers were not fully resolved, and the paper ultimately falls short of the bar for ICML. First, while the proposed framework is well-motivated and empirically validated, the core methodological contributions are limited. The combination of knowledge distillation, hypergraph modeling, and contrastive learning, though novel in this particular composition, largely repurposes established ideas without sufficiently deep theoretical or algorithmic innovation. This was particularly noted by Reviewer zAKq, who pointed out that components such as contrastive distillation and multimodal fusion are already well-explored in the literature.

Second, although the rebuttal addressed several technical inaccuracies and expanded on the theoretical aspects, these improvements came late in the process and were not acknowledged by multiple reviewers. Furthermore, the reliance on a strong teacher model that uses an LLM raises fairness issues in comparative evaluation, as several baseline models do not incorporate similar textual information. The authors’ clarification that DeBERTa is only used offline and on short reviews does mitigate this concern somewhat, but it remains unclear how generalizable the results are across deployment settings.

Third, the clarity and rigor of the paper could be further improved. Issues with notation, the mismatch between loss functions and evaluation metrics, and the inconsistent description of the target task (implicit vs. explicit feedback) suggest that parts of the submission were not polished to a high standard. While the authors made commendable efforts to clarify these during the rebuttal, the responses came too late to significantly alter the overall evaluation trajectory.